# Chromatin compaction precedes apoptosis in developing neurons

Renata Rose[1], Nicolas Peschke [1], Elena Nigi [1], Márton Golléri [2], Sandra Ritz[2], Christoph Cremer[2,3,4], Heiko J. Luhmann [1✉] & Anne Sinning [1✉]

While major changes in cellular morphology during apoptosis have been well described, the subcellular changes in nuclear architecture involved in this process remain poorly understood. Imaging of nucleosomes in cortical neurons in vitro before and during apoptosis revealed that chromatin compaction precedes the activation of caspase-3 and nucleus shrinkage. While this early chromatin compaction remained unaffected by pharmacological blockade of the final execution of apoptosis through caspase-3 inhibition, interfering with the chromatin dynamics by modulation of actomyosin activity prevented apoptosis, but resulted in necrotic-like cell death instead. With super-resolution imaging at different phases of apoptosis in vitro and in vivo, we demonstrate that chromatin compaction occurs progressively and can be classified into five stages. In conclusion, we show that compaction of chromatin in the neuronal nucleus precedes apoptosis execution. These early changes in chromatin structure critically affect apoptotic cell death and are not part of the final execution of the apoptotic process in developing cortical neurons.

[1] Institute of Physiology, University Medical Center of the Johannes Gutenberg University, Mainz, Germany. [2] Institute of Molecular Biology gGmbH (IMB), Mainz, Germany. [3] Kirchhoff Institute for Physics (KIP), University of Heidelberg, Heidelberg, Germany. [4] Interdisciplinary Center for Scientific Computing (IWR), University of Heidelberg, Heidelberg, Germany. ✉email: luhmann@uni-mainz.de; asinning@uni-mainz.de

Apoptosis is an important and thus tightly regulated process described in many if not all cell types investigated[1]. During development of the central nervous system in mammals, apoptosis is essential for the proper structural and functional maturation of the brain[2]. At the cellular level, the apoptosis process is characterized by distinct morphological changes, consisting of membrane blebbing, nuclear shrinkage, and the formation of apoptotic bodies[3–5]. Although chromatin compaction is one of these well-described morphological changes during apoptosis, descriptions of subcellular changes within the nucleus during apoptosis are limited[6–9].

With the recent technical advances such as super-resolution microscopy (SRM) and chromatin conformation capture (i.e., Hi-C/3 C) techniques, it is now possible to investigate the relationship between structure and function of the chromatin inside the nucleus[10–12]. In parallel, models seeking to explain the functional nuclear architecture have been presented[13–16]. Today, it is largely agreed that chromosome territories exist, and regions with active and inactive genes reside in different places inside the nucleus[17]. The importance of the dynamic organization of the nuclear structure, and especially chromatin, has mostly been elucidated in cell lines[18–20]. Recently, we could show that the dynamic functional organization of chromatin and plastic remodeling at the nucleus-scale are linked to activity-induced changes in gene expression in cortical neurons in vitro and in vivo[21]. Yet, a comprehensive description of structural changes of chromatin within the nucleus in neurons is still missing, and the functional significance of these subcellular changes for the execution of the apoptotic process remains unclear.

The tight regulation of apoptosis in the brain is required to limit the apoptotic removal of cells to superfluous, non-active neurons[22,23]. Thus, during the process of neuronal cell death not only a large number of genes is affected[24–26] but apoptotic degradation and removal of redundant cells in the developing brain are also possibly preceded by manifold changes in gene expression and, consequently, can be predicted by a dynamic remodeling at the structural level.

To investigate early structural changes in chromatin that may precede the classically described morphological changes during apoptosis, we used high-resolution confocal imaging, SRM namely single molecule localization microscopy (SMLM), and pharmacological tools in developing cortical neurons in vitro and in vivo. Both high resolution confocal and SRM imaging allow the structural investigation of nuclear chromatin compaction at different resolution depths, in form of changes in histone or DNA-associated fluorescence signals or in density of blinking events, respectively. Appropriate chromatin density is a basic requirement which provides the steric conditions for mobility, and thus accessibility of macromolecules and macromolecular aggregates such as individual transcription factors, transcription factor complexes or functional protein/RNA ensembles. Accessibility of individual macromolecules and macromolecular aggregates to specific intranuclear chromatin domains is a major mechanism contributing to the functional behavior of eukaryotic genomes. The initiation of transcription, for example, requires the interaction between transcription factors and target DNA sequences. The same is true for other essential nuclear functions, e.g., for replication or repair. Differential accessibility of chromatin ensembles located in the active and inactive nuclear compartments is suggested as an additional level of spatial fine-tuning of transcriptional control, as well as of other nuclear processes, such as chromatin remodeling, replication, and repair. Thus, the level of compaction of chromatin is related to silencing/activation of chromosomal territories[27,28] and has also experimentally been shown to influence gene expression under different physiological and pathological conditions[14,16,29]. But, structural

rearrangements without chromatin condensation have also been described previously e.g., the formation of senescence-associated heterochromatic foci[30]. Thus, structural changes in chromatin such as an increase in chromatin compaction have to be interpreted with caution and cannot be directly linked to changes in chromatin accessibility. For quantification of structural compaction and condensation of chromatin, Sobel edge detection algorithms can be used, since compaction and condensation of chromatin both increase the number of distinct spaces defined as edges within the nucleus[27].

The results allowed us to define and characterize five stages of chromatin organization before and during apoptosis in the nucleus down to the single-molecule scale. Here, the sequential combination of live cell confocal imaging and SRM in fixed neurons allowed us to quantify the level of chromatin compaction and clustering in each stage. Compaction of chromatin before and during this type of programmed cell death occurred progressively. Notably, time-lapse imaging also revealed an early stage of chromatin compaction that preceded the activation of classical apoptotic key players such as caspase-3 and major changes in nuclear morphology, i.e., cell shrinkage and nuclear fragmentation. SRM imaging of apoptotic vs non-apoptotic neurons in cortical tissue suggested that different stages in chromatin compaction were also present in cortical neurons during apoptosis in vivo. Results from pharmacological interference experiments further show that actomyosin activity and changes in nuclear myosin IC expression during the progression of staurosporine-induced apoptosis modulate chromatin dynamics, suggesting that early changes in chromatin are necessary for the execution of apoptotic cell death in neurons. However, our results also show that these early changes are not affected by blockade of the final execution of the apoptotic process per se. Considering the importance of apoptosis in neuronal development, characterizing these early changes in nuclear morphology is crucial for the understanding of the whole apoptosis process.

## Results

**Early phase of apoptosis is characterized by high chromatin compaction.** The first aim of the present study was to investigate the chromatin dynamics in primary neurons undergoing apoptotic cell death (Fig. 1a) by live-cell, spinning disk confocal microscopy of the histone H2B fused to the mCherry fluorescent signal.

No difference in viability was detected in H2B::mCherry overexpressing neurons when compared to non-transduced neurons, neither under non-treated control conditions (Suppl. Fig. 1, post-hoc test $p = 0.99$ for control no H2B::mCherry vs. control H2B::mCherry) nor under staurosporine treatment (post-hoc test $p = 0.95$ for staurosporine-treated no H2B::mCherry vs. staurosporine-treated H2B::mCherry). In agreement with previous studies[21,31], H2B::mCherry-positive neurons displayed a normal morphology and function (Fig. 1a).

Changes in H2B::mCherry signal densities within the nucleus of transduced neurons reflect changes in either chromatin compaction or condensation, since both are generally associated with an increase of spaces[27]. Consistent with an increase in chromatin condensation, H2B::mCherry fluorescence within nuclei of neurons was strongly co-localized with the heterochromatin marker trimethyl-Histone H3 (Lys27)[32], both under control conditions and during the apoptotic process (Suppl. Fig. 2, Pearson's Coefficient: control $0.93 \pm 0.004$ apoptosis $0.86 \pm 0.02$, $n = 5/7$ cells). Yet, it cannot be excluded that the observed structural changes in chromatin structure during apoptosis are at least partly attributed to structural rearrangement without chromatin condensation as described previously e.g., in senescent cells[30]. Thus, we refer to the increase in H2B::mCherry

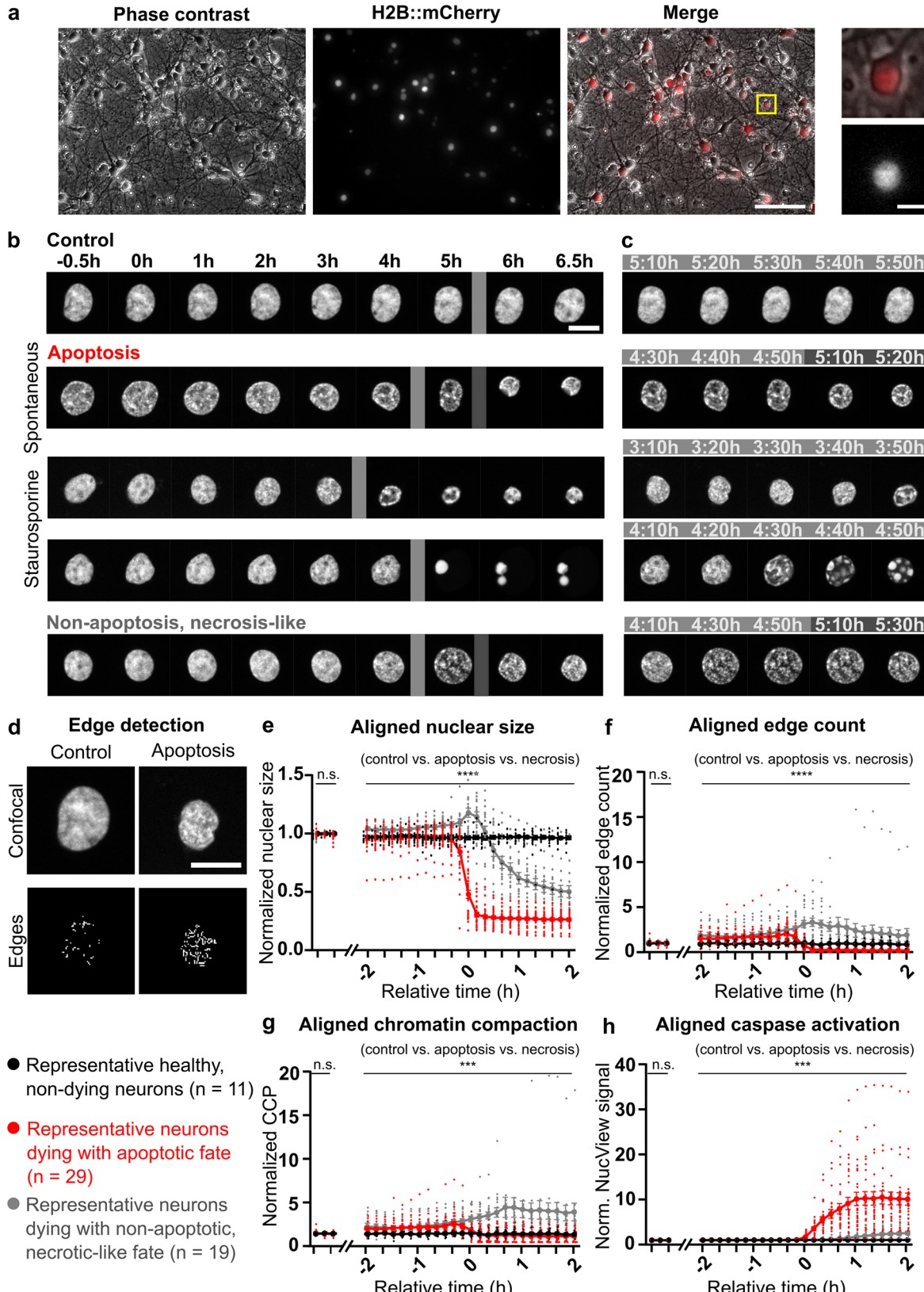

signal densities as chromatin compaction. Primary neurons under control conditions showed no change in nuclear size and only minor changes in chromatin organization (Fig. 1b, c, SupMov. 1), indicating that nuclear chromatin can move without changing the internal architecture and thus, confirming the dynamics of the nucleus[21,33,34].

In spontaneously dying neurons or upon pharmacological induction of apoptotic cell death with the known protein kinase inhibitor and inducer of apoptosis staurosporine[35,36], the previously described nuclear compaction occurring during the late phase of apoptosis was clearly visible, i.e., nuclear compaction and nuclear collapse[6,7,37]. However, high resolution imaging

**Fig. 1 Nuclear structure and chromatin dynamics during apoptotic and non-apoptotic, necrotic-like cell death in primary cortical neurons.**
**a** Representative primary cortical culture at DIV 8 transduced with recombinant adeno-associated virus (AAV) to express the chromatin marker H2B::mCherry under the human synapsin promotor; left: phase contrast, middle: H2B::mCherry, right: merge; scale 100 µm. Detail on the far right: top: merged, bottom: H2B::mCherry; scale 10 µm. **b** Hourly captures of representative cells during a 7 h time lapse acquisition, before and after start of apoptotic or non-apoptotic processes, scale 10 µm. Gray vertical lines indicate which time points are depicted in **c**, see **SuppMov.1-5**. **c** Depiction of chromatin dynamics during the most striking nuclear changes for typical types of cell death between time points indicated with the gray vertical lines in **b**. **d** Typical examples for raw confocal signal (top) and Sobel-filter-based edge detection (bottom) under control conditions (left) and apoptotic neurons (right), scale 10 µm. **e–h** Aligned and normalized average nuclear size, edge count, chromatin compaction parameter (CCP), and caspase activity normalized to baseline for representative neurons surviving under control conditions (black, $n = 11$), classified as apoptotic (red, $n = 29$) and classified as non-apoptotic, necrotic-like (gray, $n = 19$). Corresponding signals were aligned according to the occurrence of a change in nuclear size ($t = 0$ h is time of shrinkage for apoptosis and swelling for non-apoptosis, necrosis-like cell death). Imaging was performed every 10 min and the first 3 time points correspond to baseline. Data are represented as mean ± SEM. For **e–h** mixed-effects analyses for differences between cells with apoptotic vs necrotic cell fate vs. control conditions were applied during baseline: nuclear size $F_{(2, 168)} = 1.51e^{-15}$, $p > 0.9999$; edge count $F_{(2, 168)} = 5.48e^{-16}$, $p > 0.9999$; CCP $F_{(2, 168)} = 3.53e^{-16}$, $p > 0.9999$; caspase activity $F_{(2, 168)} = 1.01e^{-11}$, $p > 0.9999$, and for the aligned time period of −2 to +2 h: nuclear size $F_{(2, 56)} = 107.7$, $p < 0.0001$; edge count $F_{(2, 56)} = 10.82$, $p = 0.0001$; CCP $F_{(2, 56)} = 8.87$, $p = 0.001$; caspase activity $F_{(2, 56)} = 23.33$, $p < 0.0001$.

revealed that chromatin already underwent dynamic changes, which can be visually described as chromatin granulation, before these prominent nuclear changes appeared (Fig. 1c). Under control conditions and different pharmacological treatments, neurons also occasionally underwent non-apoptotic cell death that was characterized by a substantial nucleus swelling, and thus considered as non-apoptotic, necrotic-like.

To quantify the level of compaction of chromatin of single neurons before and during cell death, we first performed Sobel edge detection on the H2B::mCherry signal to longitudinally assess the number of chromatin-associated fluorescence signals as edges within the nucleus (i.e., edge count). Then, we determined the density of edges within the nucleus by normalization of the detected edges to the cross-section area (i.e., edge count/nuclear size; see also Fig. 1d and Methods). Thereby a direct measure for the level of chromatin compaction was calculated which was previously defined as chromatin condensation parameter[27] and here referred to as chromatin compaction parameter (CCP). Nuclear size, edge count and CCP were then compared across neurons undergoing no cell death (control), apoptotic, or necrotic-like cell death (Fig. 1e–g). Although more types of cell death have been described[38], for clarity we only classified and analyzed cells with an apoptotic or non-apoptotic, necrotic-like cell fate. The assignment of neurons to the respective cell fate was confirmed experimentally by the application of the caspase substrate NucView, which allows real-time apoptosis detection[39] (Fig. 1h). Nuclear shrinkage is a typical hallmark for neurons with an apoptotic fate, whereas non-apoptotic, necrotic-like cell death is characterized by nuclear swelling[4,40]. Thus, nuclear shrinkage or swelling are hallmark features of the respective cell death fates and both processes affect cellular morphology. To quantitatively compare onset, course and sequence of changes in chromatin structure (e.g., early vs late changes), time-lapse imaging results of representative neurons were temporally aligned according to the occurrence of a change in nuclear size (i.e., relative time $t = 0$ h, with nuclear size decreasing for apoptosis and increasing for non-apoptosis). Average values 2 h before and after nuclear size changes occurred were compared across cells with different fates (Fig. 1e–h).

Control cells did not show major changes in nuclear size, chromatin structure or caspase activation (−2 h and +2 h nuclear size: 0.97 ± 0.01 and 0.96 ± 0.02; edge count: 0.92 ± 0.11 and 0.87 ± 0.08; CCP: 0.94 ± 0.11 and 0.90 ± 0.08; NucView signal: 1.02 ± 0.004 and 1.06 ± 0.01, Fig. 1e–h, SupMov. 1). In addition to control and apoptotic cells, neurons dying via necrotic-like cell death were also analyzed (Fig. 1b). As expected, major morphological changes during necrosis differed from those observed during apoptosis[40], i.e., nuclear swelling in necrotic neurons vs. nuclear shrinkage in apoptotic neurons. Beside these clear differences in

appearance and dynamics during the execution phase of the different types of cell death, earlier subtle changes in chromatin morphology were visible in both types of cell death (Fig. 1b, c and SupMov. 2-5). The relationship between nuclear size, chromatin compaction (edge count and CCP), and caspase activation differed between apoptotic or necrotic nuclei (mixed-effects analysis for differences between cells under control conditions, apoptotis, and necrosis-like cell death: nuclear size: $F_{(2, 56)} = 107.7$, $p < 0.0001$; edge count: $F_{(2,56)} = 10.82$, $p = 0.0001$; CCP: $F_{(2,56)} = 8.87$, $p = 0.0005$; and caspase activation: $F_{(2,56)} = 23.33$, $p < 0.0001$, Fig. 1e–h).

As expected, apoptotic neurons showed a shrinkage of their nuclear size (0.97 ± 0.02 at −2 h and 0.26 ± 0.01 at +2 h; post-hoc test at +2 h $p < 0.0001$ for control vs. apoptosis), which was not preceded by any swelling (Fig. 1e). The relative nuclear size for apoptotic neurons decreased sharply within 30 min, until it reached about one-third of the control size (0.94 ± 0.02 at −10 min and 0.29 ± 0.02 at +20 min, Fig. 1e). Necrotic-like neurons, on the other hand, showed a significant increase in nuclear size that peaked at +10 min (1.03 ± 0.02 at −2h and 1.17 ± 0.04 at +10 min; post-hoc test at +10 min $p = 0.003$ for control vs. necrosis) before going through a final phase of nuclear shrinkage (relative nuclear size 0.51 ± 0.05 at +2 h; post-hoc test at +2 h $p < 0.0001$ for control vs. necrosis, Fig. 1e). With the alignment of the time courses to the major changes in nuclear size, it became clear that in apoptotic neurons the average chromatin compaction was already higher at least 2 h before a significant decrease in nucleus size occurred (−2h edge count 1.51 ± 0.12, post-hoc test $p = 0.003$, and relative CCP 1.55 ± 0.12, post-hoc test $p = 0.002$ for control vs. apoptosis, Fig. 1f, g).

For necrotic neurons, a significant increase in both edge count and CCP was apparent at least 50 min prior to the peak in nuclear size (−40 min edge count 2.02 ± 0.32, post-hoc test $p = 0.01$; CCP 1.92 ± 0.30, post-hoc test $p = 0.02$ for control vs. necrosis, Fig. 1f, g). However, in addition to the differences in time-dependent changes in nuclear size (swelling followed by shrinkage), necrotic nuclei also showed a much stronger secondary increase in edge count and CCP that remained present after the start of the shrinkage at +20 min (edge count 0.29 ± 0.05 for apoptosis and 3.35 ± 0.45 for necrosis; post-hoc test $p < 0.0001$ for apoptosis vs. necrosis; CCP apoptosis 0.89 ± 0.14 and necrosis 2.88 ± 0.38; post-hoc test $p < 0.0001$ for apoptosis vs. necrosis, Fig. 1f, g). While the nuclear size decreased, edge count slightly decreased and CCP slightly increased (−10 min and +2 h edge count 2.46 ± 0.32 and 1.84 ± 0.65; CCP 2.31 ± 0.31 and 3.35 ± 0.89, Fig. 1f, g), as a result of heterogeneous regions of high and low chromatin compaction formed during the necrotic death, which were preserved during the shrinkage.

In non-apoptotic, necrotic-like cells only a slight increase in NucView signal was detected during the late phase of cell death (Fig. 1h). This minor increase can be attributed to passive diffusion and DNA-binding of the dye, and thus does not indicate relevant caspase activation. For apoptotic neurons, the number of detected edges and CCP decreased with the shrinkage of the nucleus as a result of the formation of highly condensed and homogeneous pyknotic nuclei ($-10$ min and 2 h edge count $2.12 \pm 0.27$ and $0.22 \pm 0.05$; CCP $2.19 \pm 0.25$ and $0.73 \pm 0.13$).

In summary, high resolution imaging experiments of the dynamics of nuclear chromatin in living transgenic cortical neurons in vitro revealed that dynamic reorganization of chromatin precedes a change in nuclear size and major changes in nuclear morphology during apoptosis. Towards the end of the cell death process, nuclei from neurons dying via apoptosis became homogenous with a different chromatin organization from that of necrotic nuclei. These results show the early compaction of chromatin and the differences in chromatin rearrangement dynamics for these two types of neuronal cell death.

**Inhibition of caspase-3 and actomyosin disrupts the apoptotic process.** Given the temporal offset of the chromatin reorganization, our next aim was to investigate whether the early chromatin compaction was necessary for the execution of apoptosis and/or if this reorganization of the chromatin is part of the destruction of the neuron during cell death. For this purpose, we performed high resolution imaging experiments, in which we either disrupted the progression and final execution of staurosporine-induced apoptosis by concomitant application of the caspase-3 inhibitor Z-DEVD-FMK, or manipulated chromatin dynamics by application of 2,3-butanedione monoxime (BDM). The detection of caspase-3 activity and thus of apoptotic events was enabled by the presence of cell-membrane permeable fluorogenic caspase substrate NucView. Here, mean values for nuclear size, edge count and CCP across neuronal cultures were analyzed without temporal alignment of single cell behavior (as opposed to in Fig. 1) to compare the effect of pharmacological treatments on the cultured neurons (Fig. 2). Next, all neuronal nuclei under the different pharmacological treatments were classified into four categories (a) no change, (b) apoptosis, (c) non-apoptosis, necrosis-like, and (d) granulation reflecting chromatin compaction, based on caspase activity, nuclear size and chromatin structure, to depict the effect of treatments on chromatin dynamics and cell survival (Fig. 3, see Methods).

Under untreated control conditions, nuclear size, edge count and CCP remained stable throughout the time course of experiments (Fig. 2a–c), and only a low percentage of apoptotic events was detected based on caspase activity ($0.22 \pm 0.22\%$, $n = 9$ cultures, Fig. 2d). The overall rate of dead cells identified based on the categorical classification of neuronal nuclei, i.e., NucView signal, nuclear size and nuclear structure, was rather low (6/460 cells from 9 cultures, i.e., 1.3%, Fig. 3a). In cultures treated with staurosporine ($1.5 \, \mu M$), the average nuclear size of neurons decreased as a consequence of the shrinkage of the nuclei from dying cells, while edge count and CCP increased as a result of the chromatin compaction (Fig. 2a–d). A significant increase in both parameters, average edge count and CCP, could be detected as early as 40 min after the application of staurosporine (post-hoc test for edge count and CCP at 40 min $p = 0.01$ and $p = 0.01$ for control vs. staurosporine). In agreement with the results from the analysis of temporally aligned single neuron behavior (Fig. 1), a significant increase in NucView signal and a decrease in nuclear size was, however, only detectable after 240 min (post-hoc test for % of NucView-positive cells and nuclear size at 240 min $p = 0.02$

and $p = 0.02$ for control vs. staurosporine). As expected, the application of staurosporine presented a higher percentage of neurons displaying an apoptotic event, identified based on a strong and transient elevation of the NucView signal (26/510 cells from 9 cultures, i.e., $5.1 \pm 1.12\%$, Fig. 2d). Also, a high total number of apoptotic cells was identified based on the categorical classification (58/518 cells from 9 cultures, i.e., 11.2%, Fig. 3b). On the population level, the application of staurosporine induced a strong activation of caspases during the investigated time course of 6 h (control $100 \pm 0.91\%$, 3 h staurosporine $215.9 \pm 18.48\%$ of control, 6 h staurosporine $326.3 \pm 38.76\%$ of control post-hoc test $p = 0.03$ for control vs. 3 h staurosporine, and $p = 0.0005$ control vs. 6 h staurosporine, $n = 10$ cultures each, Fig. 2e).

To disrupt the progression and block the final execution of staurosporine-induced apoptosis, we applied the caspase-3 inhibitor Z-DEVD-FMK ($100 \, \mu M$)[41] together with staurosporine. Under these conditions and unlike the staurosporine-only treated neurons, no cortical neurons showed an apoptotic fate, as quantified based on the transient elevation of the NucView signal (NucView-positive cells 0%, Fig. 2d) and an overall low percentage of apoptotic neurons based on the categorical classification (2/265 cells from 5 cultures, i.e., 0.8%, Fig. 3c). The inhibition of caspase activity under concomitant application of Z-DEVD-FMK with staurosporine was confirmed by an almost absolute reduction of caspase activity in the luminescent caspase assay (3 h staurosporine with caspase inhibitor $6.17 \pm 1.31\%$ of control, and 6 h staurosporine with caspase inhibitor $6.12 \pm 0.42\%$ of control, Dunn's multiple comparisons test for 3 h staurosporine vs. 3 h staurosporine with caspase inhibitor $p = 0.008$, and for 6 h staurosporine vs. 6 h staurosporine with caspase inhibitor $p = 0.0003$, Fig. 2e). Thus, the simultaneous application of Z-DEVD-FMK effectively blocked the execution of apoptosis under staurosporine treatment. The quantitative analysis of chromatin compaction upon staurosporine treatment in presence of the caspase-3 inhibitor revealed, that neurons still presented a significant increase in edge count and CCP (0 h and $+6.5$ h edge count $1.04 \pm 0.02$ and $1.96 \pm 0.1$; CCP $1.05 \pm 0.02$ and $2.1 \pm 0.1$, Fig. 2b, c), which was similar in its time course and extent to the chromatin compaction observed in staurosporine-only treated cultures (edge count $1.06 \pm 0.02$ and $1.87 \pm 0.07$; CCP $1.06 \pm 0.02$ and $2.03 \pm 0.07$). Thus, no significant differences in time course and extent of chromatin compaction were found between these two treatments (post-hoc tests 0–6.5 h edge count $p \geq 0.35$ and relative CCP $p \geq 0.22$ for staurosporine vs. caspase-3 inhibitor with staurosporine, Fig. 2b, c). However, quantitative analysis also showed that for neurons treated with caspase-3 inhibitor in combination with staurosporine, nuclear size remained stable throughout most of the window of investigation, with a slight, yet significant decrease in mean nuclear size occurring only in the last 30 min (0 h, $+6$ h and $+6.5$ h nuclear size: $0.99 \pm 0.001$, $0.94 \pm 0.01$ and $0.92 \pm 0.02$; post-hoc test $p = 0.99$, $p = 0.03$ and $p = 0.02$ for control vs. caspase-3 inhibitor with staurosporine, Fig. 2a).

In summary, treatment with caspase-3 inhibitor interfered with the execution of apoptosis (i.e., caspase activity, nuclear size change and overall apoptotic rate), but not with the preceding chromatin compaction (i.e., immediate increase in edge count and CCP upon staurosporine treatment). This indicates not only that the early increase in chromatin compaction, which temporally precedes caspase activation and nuclear shrinkage, antecedes caspase-3 activation and the final execution of the apoptotic process; but that it can also be experimentally decoupled from the final apoptotic phase e.g., by pharmacological block of the apoptotic key player caspase-3. Thus, we reason that these early changes in chromatin compaction are not part of the final execution of the apoptotic process per se.

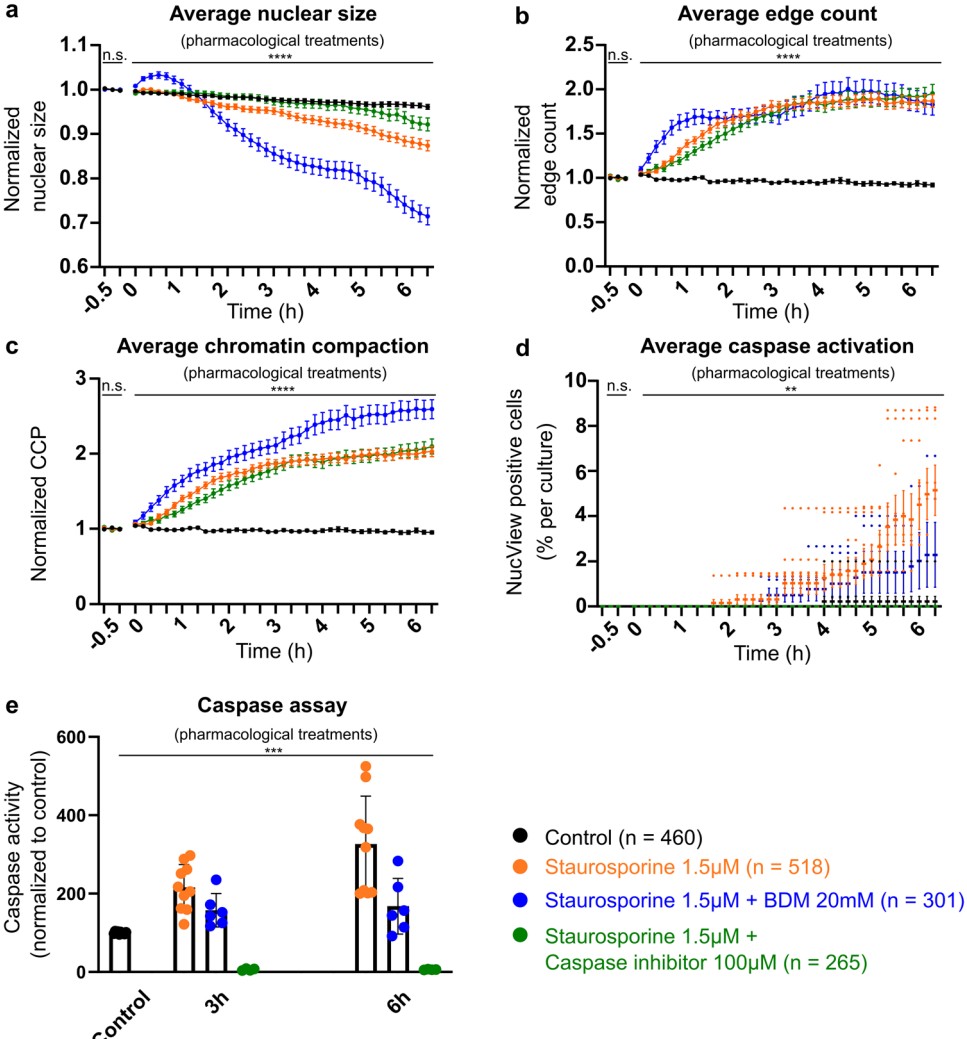

**Fig. 2 Pharmacological inhibition of caspase-3 blocked apoptosis but not preceding compaction, whereas blockade of myosin activity induced non-apoptosis, necrosis-like cell death.** Live cell confocal images were acquired every 10 min for 3 time points before the application of the pharmacological treatments (baseline), and for an additional 39 time points (i.e., 6.5 h) after application of indicated pharmacological treatments. **a–c** Normalized nuclear size, edge count and CCP for all neurons under control (black, $n = 460$ cells), staurosporine (1.5 µM; orange, $n = 518$ cells), BDM (20 mM) with staurosporine (blue, $n = 301$ cells), and caspase-3 inhibitor Z-DEVD-FMK (100 µM) with staurosporine (green, $n = 265$ cells) conditions. Note, that while the combined application of caspase-3 inhibitor with staurosporine blocked average caspase activation and reduced the nuclear size change, no significant differences in time course and extent of early chromatin compaction were found between caspase-3 inhibitor with staurosporine and staurosporine-only treated neurons. **d** Mean percentage of NucView-positive cells (caspase activation) per culture (control $n = 9$ cultures, staurosporine $n = 9$ cultures, caspase-3 inhibitor Z-DEVD-FMK with staurosporine $n = 5$ cultures, BDM with staurosporine $n = 5$ cultures) identified throughout the acquisition. **e** Caspase activity measured by luminescent caspase assay at DIV8 under baseline conditions, as well as 3 h and 6 h after start of pharmacological treatments. Data are represented as mean ± SEM. For **a–d** two-way ANOVA was applied to detect differences across pharmacological treatments during baseline: nuclear size $F_{(3, 4620)} = 0$, $p > 0.9999$; edge count $F_{(3, 4620)} = 3.45e^{-13}$, $p > 0.9999$; CCP $F_{(3, 4620)} = 0.0$, $p > 0.9999$; mean percentage of NucView-positive cells $F_{(3, 24)} = 0.000$, and during the time period after application of pharmacological treatments: nuclear size $F_{(3, 60060)} = 906.2$, $p < 0.0001$; edge count $F_{(3, 60060)} = 1526$, $p < 0.0001$; CCP $F_{(3, 60060)} = 2344$, $p < 0.0001$; mean percentage of NucView-positive cells $F_{(3, 24)} = 4.82$, $p = 0.0092$. For **e** a Kruskal-Wallis test was applied: $H_{(7)} = 39.56$, $p < 0.001$.

In order to investigate if the early changes in chromatin that precede apoptosis are necessary for the induction of this type of cell death, we next interfered with the dynamics of the chromatin organization by blocking actomyosin activity[21] under application of staurosporine. For this purpose, we applied BDM (20 mM), an inhibitor of myosin ATPase activity, which disrupts the nuclear actomyosin dynamics involved in the movement of nuclear bodies[42–44].

On the population level, in contrast to the above-mentioned significant increase in caspase activity upon sole staurosporine treatment, simultaneous application of BDM and staurosporine did not significantly increase caspase activity compared to control values, neither at 3 h (157.3 ± 17.45%; $n = 6$ cultures, Dunn's multiple comparisons test for 3 h staurosporine with BDM vs. control $p > 0.99$, Fig. 2e) nor at 6 h (167.8 ± 28.91%; $n = 6$ cultures, Dunn's multiple comparisons test for 6 h staurosporine with BDM vs. control $p > 0.99$, Fig. 2e). Treatment with BDM alone did not alter caspase activity and did not affect the chromatin structure, since 132 out of 164 cells (i.e., 80.49%, $n = 3$ cultures) were classified as "no change" (Fig. 3d). However, with BDM, the number of necrotic cells (14.01 ± 0.31%,) increased (unpaired t-test $p < 0.0001$) as compared to control conditions

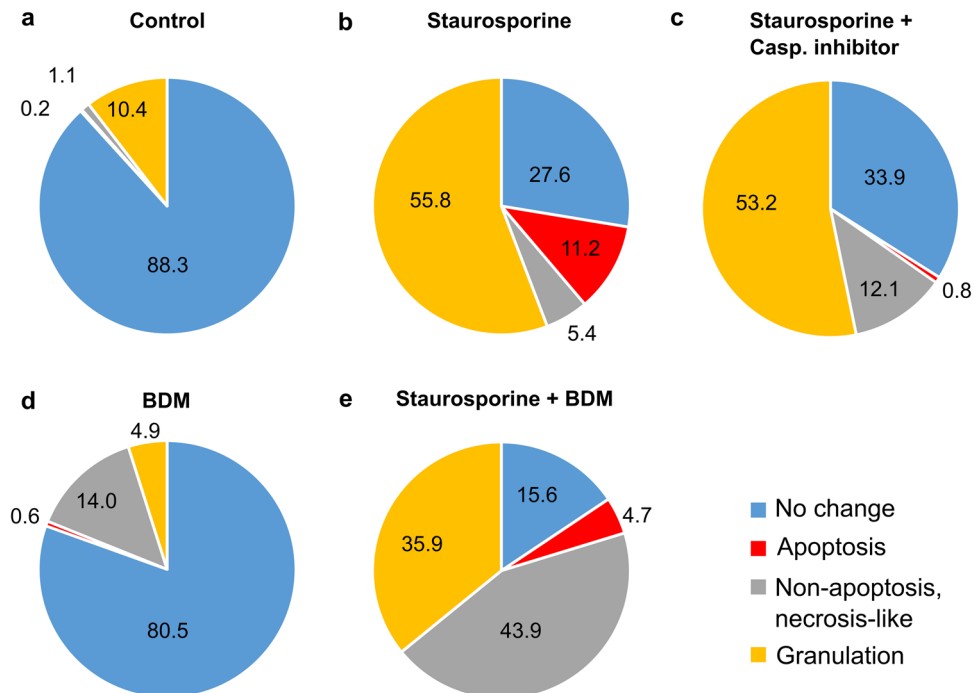

**Fig. 3 Relative proportion of cell fates following different pharmacological treatments.** Relative percentage of neurons that display no change, clear apoptotic, non-apoptotic necrotic-like cell fate, or high chromatin compaction defined as granulation upon different pharmacological treatments (control $n = 460$ cells from 9 experiments; staurosporine $n = 518$ cells from 9 experiments; staurosporine with caspase-3 inhibitor $n = 265$ cells from 5 experiments; BDM $n = 164$ cells from 3 experiments; staurosporine with BDM $n = 301$ cells from 5 experiments). **a** For most neurons under control condition, no change in nuclear appearance was observable with no significant caspase activation (measured by the intensity of the NucView signal) and no chromatin compaction until the end of 6.5 h live cell experiment (blue). **b** In staurosporine-treated cultures, many nuclei presented either a strong caspase activation and a decrease in nuclear size (apoptosis in red) or nuclei did not present a decrease in nuclear size but presented an increase in CCP, thus classified as "granulated" (yellow). **c** Whereas, if staurosporine treatment was combined with the application of the caspase-3 inhibitor Z-DEVD-FMK, only few nuclei presented an increase in caspase activation, but more neurons showed a decrease in nuclear size and a high CCP and edge count at the end of the experiment and were thus classified as necrotic (gray). **d** Under BDM treatment, an increase in the relative proportion of neuronal nuclei that presented an increase in caspase activation, a decrease in nuclear size and a high CCP and edge count at the end of the experiment, thus classified as necrotic (gray), was observed. **e** If staurosporine treatment was combined with the application of BDM, the number of neuronal nuclei classified as necrotic (gray) increased, as well as the number of granulated nuclei.

$(1.04 \pm 0.43\%$, $n = 9$ cultures, Fig. 3a, d). In cortical cultures treated with staurosporine, the additional application of BDM resulted in an immediate and unexpected induction of chromatin compaction represented by an increase in edge count and CCP as early as 20 min after application ($+20$ min edge count $1.35 \pm 0.07$; post-hoc test $p = 0.0002$; CCP $1.28 \pm 0.06$; post-hoc test $p = 0.005$ for control vs. staurosporine with BDM, Fig. 2b, c). This increase in chromatin compaction occurred earlier than in staurosporine-only treated cells, and was accompanied by a sharp significant increase in nuclear size (nuclear size $1.01 \pm 0.003$ at 0 h, and $1.03 \pm 0.01$ at 20 min; post-hoc test $p = 0.02$ for control vs. staurosporine with BDM at 20 min, Fig. 2a). The time course and extent of these changes in chromatin organization and the nuclear swelling resembled the observations in necrotic neurons (Fig. 1a, b), and thus indicate that under concomitant application of BDM, staurosporine did not induce cell death via apoptosis, but rather induced a necrosis-like death in cortical cultures. In line with that, only $2.29 \pm 1.43\%$ of cortical neurons treated with staurosporine in the presence of BDM showed an increase of the NucView signal throughout the recording (Fig. 2d), which was significantly less than under staurosporine alone ($5.15 \pm 1.12\%$, t-test $p = 0.0001$). In agreement, categorical classification of cell fate based on NucView signal and nuclear morphology also showed that under combined application of BDM and staurosporine, 43.9% of the neurons were classified as necrotic-like while only 4.7% appeared apoptotic. Thus, concomitant

application of BDM resulted in the alteration of staurosporine-induced chromatin dynamics with less apoptotic and more necrotic-like cell death. This effect on chromatin dynamics presumably is due to blockade of actomyosin activity by BDM, but unspecific side effects of BDM, such as interference with other non-nuclear processes in cells[45], cannot be excluded. Besides the conventional myosin II, at least 7 of the known unconventional myosins are described to be expressed in the nucleus albeit with largely unknown functions[46,47]. To address the role of myosin for the dynamics of chromatin organization during apoptosis, we next performed immunocytochemical stainings of the unconventional myosin IC and the conventional myosin IIA in primary cortical cultures under control conditions and upon induction of cell death by application of staurosporine, i.e., neurons in the early vs. late phase of apoptosis. The results confirmed the presence of myosin IIA and IC in the nucleus of primary cortical neurons. While we did not observe significant changes in the expression of the conventional myosin IIA during the early and late phase of apoptosis compared to control conditions (Suppl. Fig. 3), the nuclear expression of myosin IC significantly decreased with the progression of the apoptotic process (Fig. 4a, c). Interestingly, this decrease in expression of the actin-based molecular motor protein myosin IC upon application of staurosporine was significantly attenuated through concomitant application of BDM after 2 h (early phase apoptosis; Fig. 4b, d) and 6 h (late phase of apoptosis; Fig. 4b, e). In line with no change

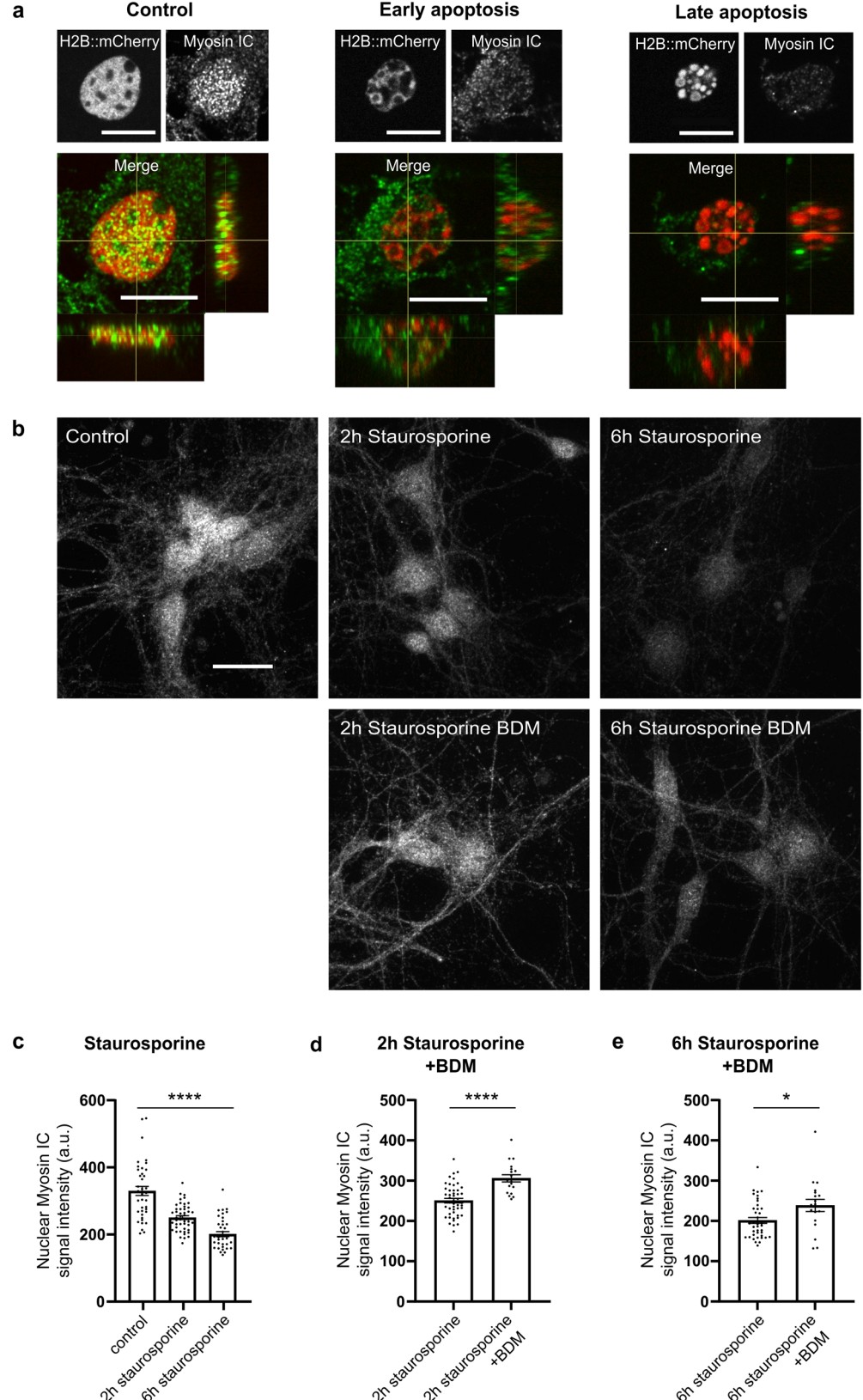

in nuclear expression levels of myosin IIA, those also remained unaffected by the BDM treatment (Suppl. Fig. 3).

To assess whether actomyosin activity in the early phase of apoptosis induction is critical for the execution of apoptosis, we applied BDM only during the first 2 h of staurosporine application and analyzed the caspase activation after an additional 4 h of only staurosporine treatment, i.e., 6 h. The presence of BDM during the early phase (0–2 h) significantly attenuated the staurosporine-induced activation of caspase (control $100 \pm 3.59\%$, staurosporine $158.20 \pm 6.86\%$, staurosporine with BDM 0–2 h $126.2 \pm 10.60\%$; $n = 6$ cultures each; post-hoc test $p = 0.02$ for staurosporine vs. staurosporine with

**Fig. 4 Expression of the unconventional myosin IC in nuclei of primary cortical neurons decreases with the onset of apoptosis, and concomitant application of the actomyosin inhibitor BDM attenuates this decrease induced by staurosporine treatment. a** Representative confocal images of immunostainings against myosin IC, respective H2B::mCherry signals and merged images with horizontal z-stack projections confirm the expression of myosin IC in the nucleus of cortical neurons with a mostly complementary expression of myosin IC and H2B::mCherry under control conditions. In the early and late phase of apoptosis nuclear myosin IC signal decreased continuously (scales 10 μm). **b** Representative average z-stack projections of myosin IC signals under untreated control conditions, 2 h and 6 h after staurosporine application and upon additional treatment with the actomyosin inhibitor BDM (scale 20 μm). **c** Nuclear myosin IC signal intensity significantly decreased upon application of staurosporine for 2 h and 6 h ($n = 33/48/33$ cells). **d, e** The concomitant application of the actomyosin inhibitor BDM significantly attenuated the staurosporine-induced decrease in nuclear myosin IC intensity, both after 2 h ($n = 48/19$ cells) and after 6 h ($n = 33/19$ cells). Data are represented as boxplots, whiskers MIN to MAX. One-way ANOVA was applied for comparison of differences between control and staurosporine 2 h and 6 h $F(2, 123) = 46.73$, $p < 0.0001$ and t-test for comparison of nuclear myosin IC signal intensity upon application of staurosporine only with staurosporine plus BDM after 2 h ($p < 0.0001$) and 6 h ($p < 0.05$), respectively.

BDM 0–2 h, $p = 0.07$ for control vs. staurosporine with BDM 0–2 h; Suppl. Fig. 4).

Taken together, concomitant blockade of actomyosin activity by BDM attenuated changes in nuclear expression of the actin-based molecular motor protein myosin IC, modulated staurosporine-induced chromatin dynamics and resulted in less apoptotic and more necrotic-like cell death. The mitigation of caspase activation when BDM is present only in the early phase suggests that changes in chromatin dynamics, which precede caspase activation are necessary for the execution of the apoptotic cell death in neurons. Further, treatment with caspase-3 inhibitor interfered with the execution of apoptosis but not with preceding chromatin compaction supporting that these early changes in chromatin compaction are not part of the final execution of the apoptotic process per se.

**Voronoi analysis of SMLM shows progressive chromatin compaction and cluster formation during apoptosis.** To investigate chromatin dynamics during neuronal apoptosis at a resolution down to the tens of nanometers[48,49], we performed super-resolution imaging with SMLM of neurons fixed at different stages during the apoptotic process upon prior confocal live cell imaging (Fig. 5a). This combination of techniques enabled us to gain detailed insight into chromatin appearance throughout all stages of the apoptotic process, and to quantify chromatin compaction during the early phase that precedes the execution of apoptotic cell death (Fig. 5b).

Classification of apoptotic stages (Fig. 5c) was first performed manually based on the morphology of neuronal nuclei during live cell confocal imaging. Here, stage 1 described the chromatin appearance in neuronal nuclei under control condition, where chromatin was homogeneously distributed and appeared smooth or only slightly rough. At this stage, the nucleus did not present any granules or areas with high chromatin compaction. At stage 2, the neuronal nucleus lost the smoothness of chromatin distribution, as it became slightly rough with few or many granules. In this stage, spots with higher chromatin compaction that precede apoptosis were first detectable. In stage 3, chromatin distribution became more heterogeneous, with both strong granules and chromatin-free regions present. In stage 4, the nucleus lost its structure, as it collapsed becoming smaller and composed of highly compacted chromatin. Finally, stage 5 was characterized by a further reduction in nuclear size and the highest level of chromatin compaction, as at this stage the cell should be ready for phagocytosis. Machine learning algorithms for pixel and object classification in ilastik[50] were used to confirm this manual classification in a supervised but automatized way. As a result, machine learning confirmed the by-eye classification of 28 out of 46 nuclei (i.e., 61%), and another 12 (i.e., 26%) with a slight deviation of one stage. For the remainder of 6 nuclei (i.e., 13%) automatized stage classification deviated by two stages from the prior manual classification. Thus, considering these reassuring results of the algorithm classification, only

nuclei of neurons for which the classification was confirmed through the machine learning were included for further SRM analysis ($n = 28$ cells from 6 cultures). In addition, classification of neurons into respective stages 1–5 was confirmed by comparison of average nuclear size, edge count, H2B::mCherry and NucView signal intensities in confocal images of neuronal nuclei categorized into the different stages (Fig. 5d–g). Nuclear size was reduced significantly in neurons at stage 4 and 5 as compared to stage 1 (post-hoc test stage 1 vs. stage 4 $p < 0.0001$ and stage 1 vs stage 5 $p < 0.0001$), edge count increased at stage 2 (post-hoc test stage 1 vs. stage 2 $p = 0.03$), H2B::mCherry signal intensity highest in cells classified to be in stage 5 (stage 5 $2.59 \pm 0.01$) and caspase activation highest in neurons classified to be in stage 4 and 5 (stage 4 $4.18 \pm 3.01$ and stage 5 $3.91 \pm 1.42$ compared to stage 1 $1.04 \pm 0.01$).

We next aimed to describe the single molecule localization during the continuous physiological process of neuronal apoptosis in super-resolution. For super-resolution quantification of the chromatin structure and compaction at the different apoptotic stages, we performed a Voronoi analysis on polar-transformed localizations obtained from the SMLM image reconstructions (Fig. 6). For analysis, Voronoi tessellation of the localizations of individual blinking events (i.e., location of chromatin-derived histone signals) within the neuronal nuclei was calculated based on the SMLM reconstruction of each stage (Fig. 6a). The comparison of distributions of log-normalized Voronoi densities grouped by apoptotic stage thus allows the systematic comparison of the density of H2B::mCherry molecules within nuclei before and during the execution of apoptosis and under control conditions. To account for different number of localizations and cells for each stage, the area under the curve was normalized to 1 and the probability density function plotted. An increase in chromatin compaction, which can be visually described as chromatin granulation and was hitherto quantified by increasing edge counts and CCP based on confocal imaging, would be thus represented by a right-shift of the peak of probability density function due to an increase in density of individual H2B::mCherry localizations.

The results showed a log-normal, most left-hand Voronoi density distribution of H2B-mCherry molecules within the nuclei of cortical neurons under control conditions, i.e., stage 1 ($-7.83 \pm 0.42 \log(nm^{-2})$, $n = 4.56 \times 10^6$ localizations from 13 nuclei), indicating no particular clustering and high homogeneity among the localizations, as expected based on confocal data (Fig. 5b). This was also detected in the radius/density distribution, where the log-normalized Voronoi densities are homogenously distributed throughout the nuclei (Fig. 6c). In neurons identified to be at stage 2, nuclei presented with an increase in Voronoi density (right shift), indicating the beginning of cluster formation ($-7.62 \pm 0.43 \log(nm^{-2})$, $n = 1.27 \times 10^6$ localizations from 6 nuclei, Fig. 6b). Unlike the homogeneous stage 1, localizations began to accumulate in loosely formed clusters in stage 2 (Fig. 5c).

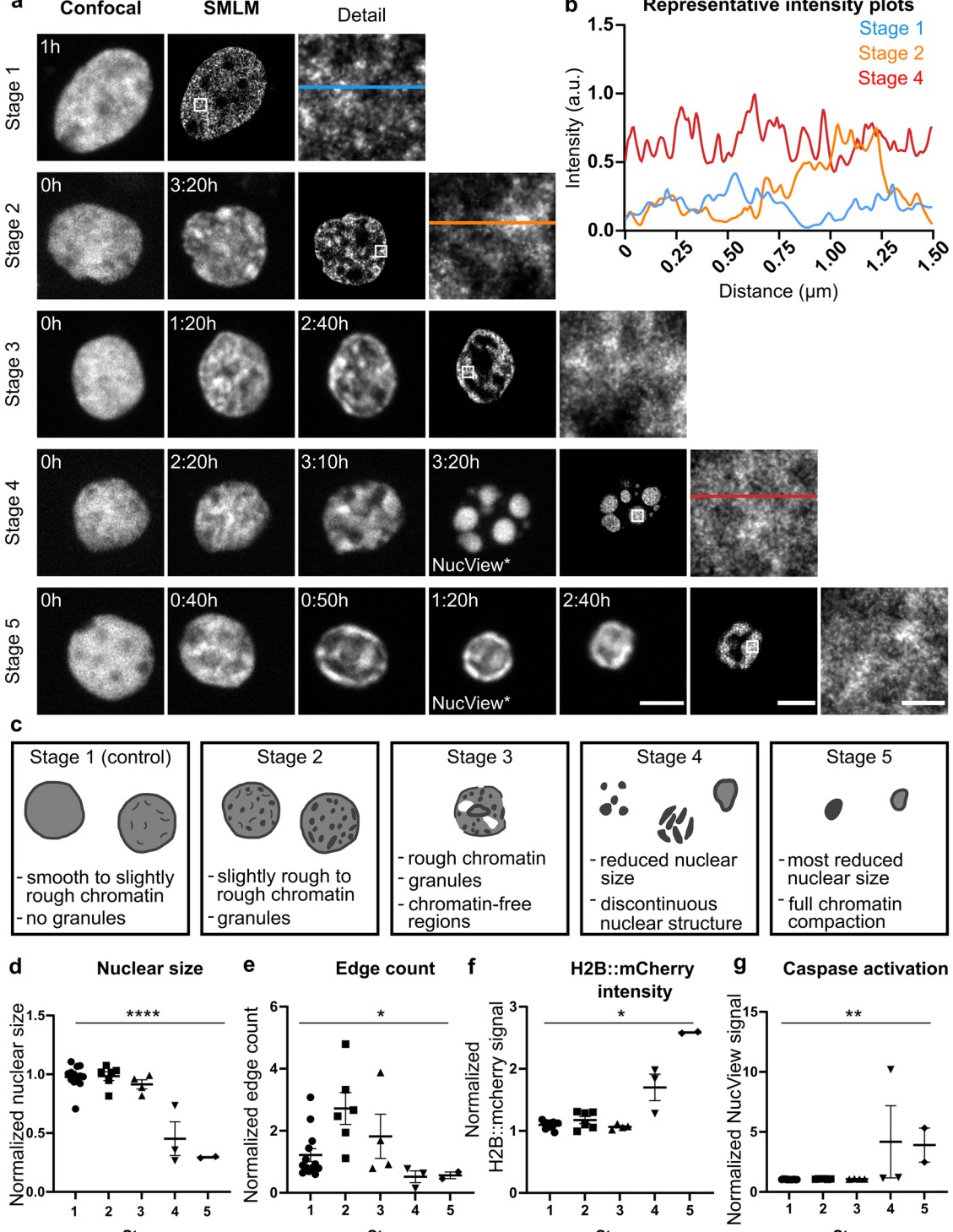

**Fig. 5 Classification of five progressive stages of apoptosis based on chromatin structure. a** Live confocal images (left, time-stamped images) followed by SMLM (images with typical details highlighted by white boxes) after fixation of the same nucleus from a cultured primary cortical neuron expressing H2B::mCherry; scale: 5 μm in both. Time stamps on the confocal images refer to time elapsed since the beginning of the acquisition. Note, in cells identified to be at stages 4 and 5, a transient caspase-3 activation (indicated by NucView*) was detectable during the confocal acquisition. Representative details highlighted in super resolution images (white boxes, scale 0.5 μm) show the compaction of chromatin increasing from stage 1 to stage 5. **b** Intensity plots of lines in representative detail views for stages 1, 2, and 4 show a stepwise increase in signal intensity, which indicates an increase in the compaction of chromatin clusters in neuronal nuclei. **c** Theoretical model describing chromatin appearance during the five identified stages of apoptosis based on the observed morphological changes presented by neuronal nuclei before and during the apoptotic process. **d–g** Quantitative comparison of average nuclear size, edge count, H2B::mCherry and NucView signal intensities of neuronal nuclei categorized into stages 1–5 ($n = 13/6/4/3/2$ cells). Data are represented as mean ± SEM. One-way ANOVA was applied for comparison of nuclear size $F_{(4, 23)}=27.02$ $p < 0.0001$ and edge count $F_{(4, 23)} = 4.11$ $p = 0.0118$, Kruskal-Wallis test for comparison of H2B::mCherry signal intensity $H(5) = 12.49$, $p = 0.0141$, and caspase activation $H(5) = 16.75$, $p = 0.002$ across nuclei at different stages.

Clusters became even denser in neurons classified as stage 3, as indicated by the higher Voronoi density, also seen by a right shift of the Voronoi density distribution $(-7.58 \pm 0.53 \log(\text{nm}^{-2})$, $n = 7.03 \times 10^5$ localizations from 4 nuclei, Fig. 6b). These findings are in line with the anticipated strong compaction of chromatin at this stage (Fig. 1b, c), and were also detected in the radius/density distribution where the log-normalized Voronoi densities were heterogeneously distributed throughout the nuclei. Here, a stronger probability of localizations was found towards the edge of the nuclei, at a radius of approximately 4000 nm distance from the center of mass (Fig. 6c). Neurons identified to be at stage 4 of apoptosis showed a wide distribution of clusters with different densities $(-7.60 \pm 0.63 \log(\text{nm}^{-2})$, $n = 1.09 \times 10^6$ localizations from 3 nuclei, Fig. 6b), representing the variety of possibilities for chromatin compaction observed at this stage (Fig. 5c). This variety was also detected by the radius/density distribution, where the nuclei lost their shape but remained compacted, preceding the final nuclear shrinkage (Fig. 6b). At last, on stage 5, homogeneous and dense clusters of chromatin were present, with a higher mean log-normalized density and narrower SD of densities as compared to all previous stages $(-7.37 \pm 0.49 \log(\text{nm}^{-2})$, $n = 4.09 \times 10^5$ localizations from 2 nuclei, Fig. 6b). The high cluster density, as well as shrinkage of the nuclei were also detected in the radius/density distribution (Fig. 6c).

In summary, these results confirmed that changes in chromatin structure, i.e., chromatin cluster formation, precede morphological changes, i.e., shrinkage of the nucleus during apoptosis, and progress continuously throughout the execution of apoptosis in developing neurons.

**High chromatin compaction in apoptotic neurons in vivo.** Developmental apoptosis and its main regulatory mechanisms are conserved across many species and experimental models[1]. To investigate and compare the chromatin structure of nuclei of apoptotic vs. non-apoptotic neurons in vivo, we performed SMLM in fixed cortical tissue of wildtype mice at postnatal day 6, i.e., during the period of increase cell death in vivo[51,52]. For this purpose, lipids were removed from tissue via incubation in CUBIC-L tissue clearing solution to avoid light refraction[53,54], and tissue was stained against activated caspase 3 (aCasp3, Fig. 7a, b). For DNA visualization, cryosections were stained with Sytox Orange, an intercalating dye (Fig. 7a, b). By treatment with an RNAse cocktail (see Methods), we eliminated the possibility of Sytox Orange to bind to RNA. For imaging of neuronal nuclei in super resolution, cryosections were mounted in fBALM blinking buffer[55], and apoptotic and non-apoptotic neurons were identified based on immunofluorescence signal for aCasp3 (Fig. 7a). Even at the peak of postnatal cell death, apoptosis affects only a minor fraction of neurons[22,56]; thus, as expected most cortical neurons were aCasp3-negative. SMLM imaging of the DNA signals of these aCasp-3-negative and -positive cells ($n = 7/6$ cells) revealed a similar pattern of fluorescence intensity as the one observed in H2B::mCherry transduced neurons (Figs. 5 and 7c). Neuronal nuclei from cryosections stained with DNA dye also showed lower fluorescence intensity for chromatin in the absence of aCasp3 signal (i.e., before apoptosis) than neurons presenting the aCasp3 signal (i.e., late phase apoptosis, Fig. 7c).

Subsequent SMLM data analysis revealed that the log-normalized Voronoi density distribution was wide and low $(7.58 \pm 0.59 \log(\text{nm}^{-2})$, $n = 2.41 \times 10^6$ localizations from 7 nuclei, Fig. 7d). This low density distribution with high variability indicated the absence of one particular cluster size and instead the presence of heterogeneous clustering among the localizations. The wide Voronoi density distribution in aCasp3-negative

neurons (Fig. 7d) suggested the presence of different sized clusters, which could be a result of higher signal intensities of chromocenters detected by intercalating Sytox Orange with their particularly high DNA content[57,58], or chromatin that was partly condensed when caspase-3 was not active yet, i.e., stage 3 in in vitro data. In opposite, aCasp3-positive neurons showed a high chromatin compaction which was significantly higher than in Casp3-negative neurons $(-7.23 \pm 0.56 \log(\text{nm}^{-2})$, $n = 1.29 \times 10^6$ total localizations from 6 nuclei, Fig. 6d). For aCasp3-positive nuclei, a narrower and higher Voronoi density distribution was detectable (Fig. 6d), a distribution of DNA signals that represented more homogeneous and denser DNA clusters which was compatible with the findings in H2B::mCherry-expressing cortical neurons in vitro during the late phase of the apoptosis process (i.e., stages 4 and 5, Fig. 6c).

Overall, nuclei of aCasp3-negative and aCasp3-positive neurons in fixed mouse cortex showed a clearly distinct distribution of chromatin clusters with a high chromatin compaction in apoptotic neurons in vivo. This was detectable by high clustering of signals of DNA-intercalating dye, which is in line with the continuous chromatin compaction during the execution of apoptotic cell death in H2B::mCherry-expressing neurons in vitro.

## Discussion

The main observations of the present study on developing cortical neurons from mice are: (i) before and during spontaneous and staurosporine-induced apoptosis, compaction of chromatin occurs progressively and can be characterized by five distinct stages; (ii) chromatin compaction precedes caspase-3 activation and major morphological changes previously described for apoptosis; (iii) the early phase chromatin compaction is functionally distinct from the latter continuous compaction of the nucleus during the apoptotic deconstruction of the neuron as it remained unaffected by caspase-3 inhibition; (iv) interfering with the dynamics of chromatin organization through inhibition of actomyosin activity resulted in inhibition of the bona fide apoptotic pathway by staurosporine. In summary, our results demonstrate that chromatin compaction is present as an essential element very early in developing neurons with a prospective apoptotic fate, but that this early reorganization of the neuronal nucleus is not part of the apoptosis process per se.

While morphological changes during apoptotic events have been described at the cellular level many years ago[3-5,8], the here observed early compaction of chromatin preceding these events in neurons had not yet been described, and is consistent with recent discoveries in studies of nuclear dynamics[18-20]. Chromatin dynamically changes its local structure because of its overall negative charge; these changes in structure possibly govern gene expression by controlling the access to DNA, consequently regulating genome readout, and thus function of the nucleus[16,59]. Thus, rearrangements in chromatin are associated with activation or interception of transcriptional programs in different physiological contexts, for example stem-cell differentiation, where increasing chromatin compaction was associated with lower pluripotency[60]. Under stressful conditions, the negative elongation factor forms nuclear condensates in human cells, which were linked to transcriptional downregulation and cellular survival[61]. Also, in the brain, 3D-genome organization was causally linked to gene-regulatory networks and chromatin conformation dynamics that interfere with brain development, function and dysfunction[62-64]. Using confocal microscopy, we have recently shown that changes in patterns of neuronal activity and specific behavioral experiences were correlated with rapid nuclear reorganization of chromatin and changes in the transcriptional program of cortical neurons[21]. In line, rapidly induced chromosome conformational changes

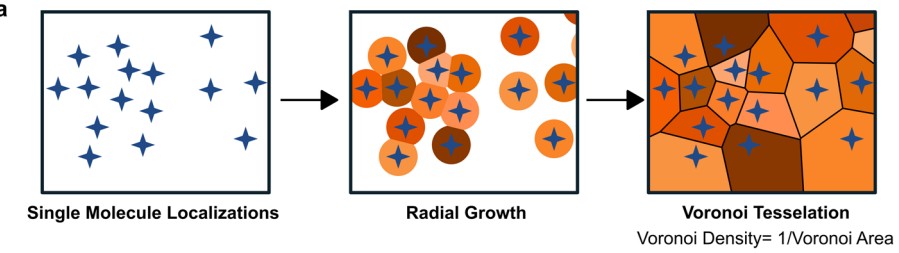

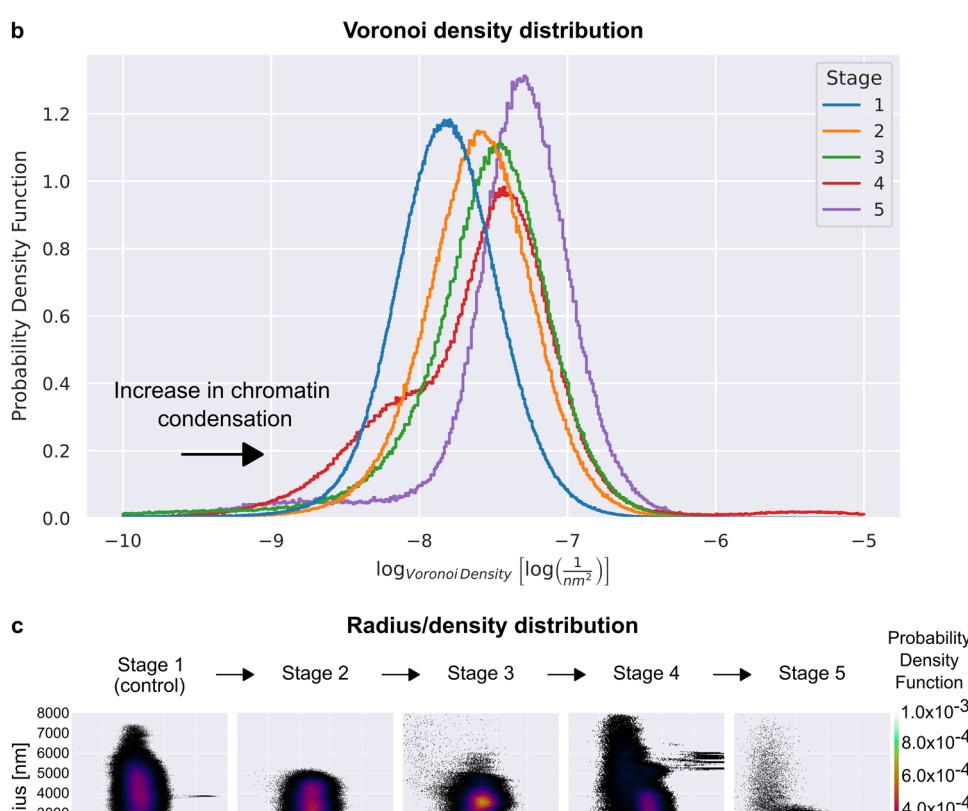

**Fig. 6 Voronoi analysis of H2B::mCherry signal localizations of super resolution data confirms a progressive compaction of chromatin before and during the apoptotic process. a** Schematic illustration of Voronoi tessellation used for calculation of Voronoi densities based on single molecule localizations. **b** Distribution of the log-normalized Voronoi densities representing the density of H2B::mCherry molecules within nuclei, grouped by apoptotic stage. To account for different number of localizations and cells for each stage, the area under the curve is normalized to 1 and the probability density function is plotted; a right-shift of the peak represents an increase in density of individual H2B::mCherry localizations, thus chromatin compaction. Stage 1: blue ($n = 13$ cells), stage 2: orange ($n = 6$ cells), stage 3: green ($n = 4$ cells), stage 4: red ($n = 3$ cells), stage 5: purple ($n = 2$ cells). **b** For the calculation of radius by density distribution, signal positions were transformed from a cartesian to a polar coordinate system for each nucleus. Respective radii from the center of mass of the nuclei were calculated for individual localizations (in nm) and related to the Voronoi density. The values of the normalized (area under curve = 1) probability density function are represented by colors (see color bar on the right).

and corresponding activation of IEGs were shown by Hi-C techniques[65]. In agreement with the importance of neuronal chromatin architecture for brain function presented in these studies, changes in chromatin organization also play a pivotal role during the early phase of apoptosis as identified here. These structural changes in chromatin likely enable the complex and rigorous regulation of this process under physiological and pathophysiological conditions, likely by affecting transcriptional activity of critical pro- and anti-apoptotic key players; however, other mechanisms such as changes in accessibility for macromolecules, interaction strength with lamina, or robustness of nuclear structure can also be affected through the global chromatin reorganization[14,16,66]. Pharmacological blockade of caspase-3, retained this early chromatin compaction with similar dynamics as staurosporine-only treated neurons. However, the bona-fide apoptotic fate was blocked, as the activation of caspase-3 was not detectable and fewer neurons died, potentially shifted towards a caspase-independent apoptotic or necrotic cell fate. Thus, this early chromatin compaction (i.e., formation of clusters or granulation) described in the present study precedes caspase activation and is distinct from the continuous condensation of the nucleus, which is a subsequence of the apoptotic destruction of the neuron.

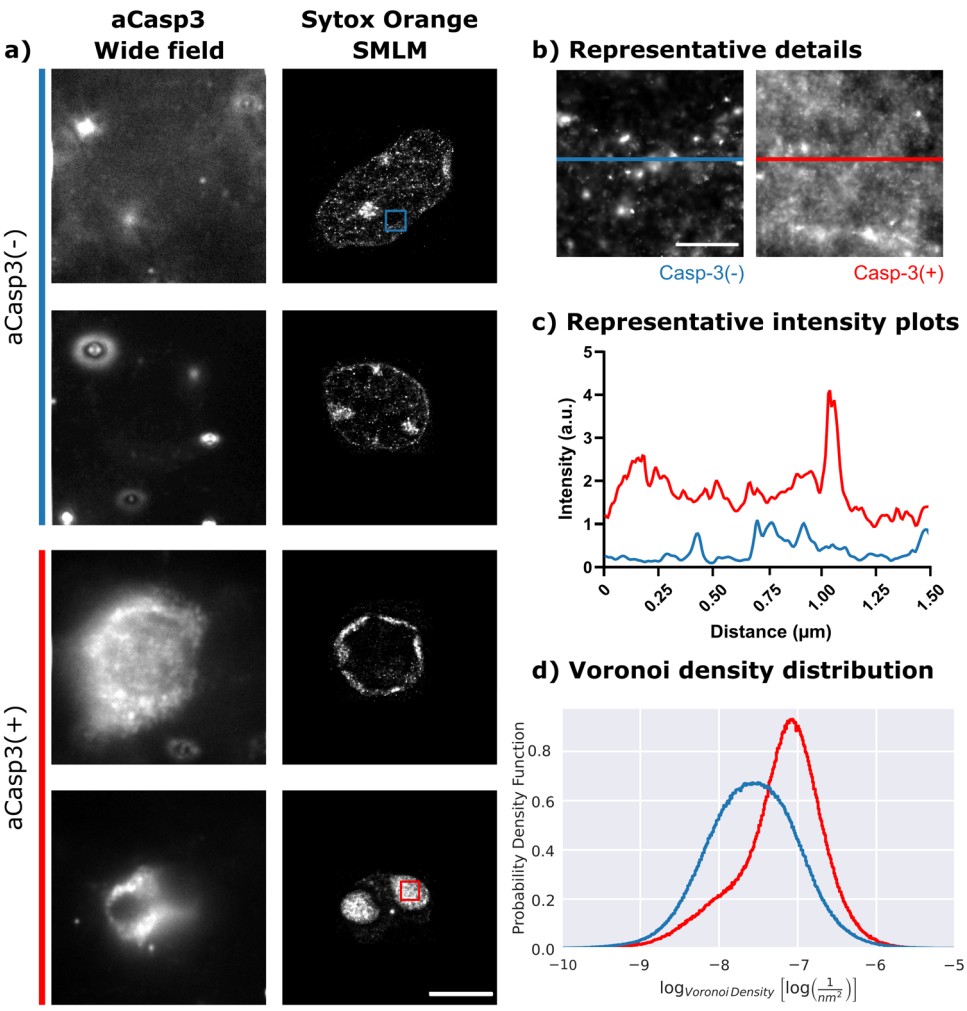

**Fig. 7 Super resolution SMLM imaging of apoptotic and non-apoptotic neurons in mouse cortical cryosections confirms high chromatin compaction during apoptosis in vivo. a** Widefield imaging of aCasp3 immunosignal (left) allowed the identification of apoptotic and non-apoptotic neurons, and SMLM reconstruction of DNA molecules stained with Sytox Orange (right) are shown for four representative cells imaged in cortical tissue samples from a P6 mouse. Scale: 5 μm. **b** Representative detailed views on inserts highlighted in **a** revealed that compaction of chromatin into clusters is higher in nuclei of neurons with aCasp3 signal (red) as compared to nuclei without caspase 3 activation (blue). Scale 0.5 μm. **c** Intensity plots of representative lines from details in **b** confirmed the higher compaction of chromatin in nuclei of neurons with the aCasp3 signal. **d** Distribution of the log-normalized Voronoi densities of aCasp3-positive (red, $n = 6$ cells) and -negative neurons (blue, $n = 7$ cells). To account for different number of localizations and cells under each condition, the area under the curve is normalized to 1 and the probability density function is plotted.

While the importance of chromatin organization is clear, the nature of chromatin condensates observed in different chromatin organization studies is still under investigation. The condensates have been shown to result from liquid-liquid phase separation (LLPS)[12,34,66], and phase separation has been proposed to underlie transcriptional regulation in super-enhancers[67], later linked to gene control in mouse embryonic stem cells[68]. More recently, however, chromatin condensates have also been postulated to have solid-like properties[69]. Independent from their physical nature, and based on the overwhelming evidence currently available, it seems to be clear that changes in chromatin compaction are in most cases associated with changes in chromatin condensation and thus affect transcription in multiple cellular processes. With few exceptions[30], changes in chromatin compaction thus likely resemble the structural correlate of transcriptional changes e.g., during the early apoptotic phase in cortical neurons as shown in the present study. In developing neurons, the prominent apoptotic pathway is believed to be the intrinsic (mitochondrial) one[70,71], where activation of transcriptional programs is involved in its execution before caspase 3 is activated. In this scenario, chromatin

compaction and condensation processes would allow for transcription of pro-apoptotic genes, whose protein products such as the pro-apoptotic pore-forming BAX and BAK from the BCL-2 family[72], lead to the initiation of cascade events and finally apoptotic deconstruction of designated neurons. In line, very similar, but transient structural changes could be directly associated with an activity-dependent increase in gene expression of immediate early genes and a transcriptional activation in a previous study[21]. However, the structural increase in chromatin compaction in apoptotic neurons could in principle also reflect a general loss of transcriptional activity. In the future, combined high resolution structural and functional analysis is needed to fully understand the functional consequences of structural changes at the molecular level. Thus, we can only speculate that the transcription of genes that lead to the execution of apoptosis is enabled by the increased chromatin condensation preceding caspase-3 activation and major changes in nuclear structure. Due to the supposed complex 3D-nanostructure of genes as predicted from detailed numerical simulations[73], chromatin condensation induced at specific nuclear sites appears to be well compatible with the importance of

enhanced spatial accessibility[16,58,74] for transcription at other sites. Changes in nuclear dynamics preceding apoptosis were also described in HeLa cells, where changes in nuclear transport factors and poly(A) + -RNA could be detected before caspase activation, and permeability increase of nuclear pore complex was happening at the same time as mRNA accumulated in the nucleus[75]. Thus, compaction of chromatin preceding apoptosis may not be exclusive to apoptosis in developing neurons.

Under concomitant application of BDM, an unspecific myosin blocker, the induction of the apoptotic process by staurosporine was blocked, the apoptosis-associated decrease in nuclear myosin IC expression was attenuated and chromatin dynamics were significantly altered. However, neurons under this combined treatment still died and displayed changes in chromatin and nuclear appearance, suggesting that cells were not dying via staurosporine-induced apoptosis but more pushed towards a necrotic cell fate. The results indicate that upon induction of apoptosis, nuclear actin-based molecular motor protein myosin IC levels decrease parallel to the observed increase in chromatin compaction. Although the low specificity of BDM and potential effects on non-nuclear myosin-dependent processes in cells warrants a cautious interpretation of these results[45], the increased expression of nuclear myosin IC and altered chromatin dynamics in parallel to the mostly necrotic fate of neurons upon BDM application indicate that early, subtler chromatin compaction is necessary for the subsequent execution of caspase-dependent apoptosis in neurons. The reduction of nuclear myosin IC upon induction of apoptotic death does not support an active role in chromatin compaction. The down-regulation of nuclear expression levels of the unconventional myosin IC, e.g., by redistribution within cellular compartments[76], might result in a destabilization of chromatin structures. This is in line with the previously described dependence of nuclear chromosome movements on nuclear myosin and actin[77–79]. Given the elusive understanding of the functional role, precise regulations and difference across the variety of myosins with a nuclear expression[46,47], the refinement of myosin IC and others' explicit roles on chromatin organization under physiological and pathophysiological conditions, such as the apoptotic process in neuronal cells, remains to be further solved in the future.

The combination of confocal live cell imaging with SRM and the classification of neuronal nuclei into consecutive stages during the apoptotic process not only revealed the different levels of chromatin compaction, but also allowed a quantitative comparison of chromatin compaction in apoptotic versus non-apoptotic neurons in intact cortical tissue. SMLM imaging data quantification with Voronoi tesselation confirmed that chromatin becomes progressively compacted from a loose configuration in control conditions (stage 1) to a slightly compacted form preceding apoptosis (stage 2) to finally a highly compacted form (stage 5) at the end of apoptosis. The detection of a similar morphology of chromatin compaction in spontaneously dying neurons during cortex development in vivo shows, that progressive compaction of chromatin is a hallmark feature of apoptotic neurons and not a consequence of the applied staurosporine treatment in vitro. This implies that a tight regulation of chromatin organization in the nucleus is not only required at the functional level before and during the apoptotic process[22,23], but also at the structural level, as chromatin has to be dynamically organized to allow the targeted elimination of superfluous, non-active neurons needed during brain development.

In conclusion, our study provides evidence for the importance of the tight regulation of chromatin organization in developing neuronal nuclei before and during apoptosis. Chromatin conformation capture techniques revealed that 3D chromatin organization is a fundamental regulatory mechanism in brain function and dysfunction[59]. Nevertheless, further investigations of chromatin organization and dynamics in neurons in the context of different physiological and pathophysiological conditions, such as apoptosis, will improve our understanding of casual relationships between spatial chromatin organization, molecular signatures, and neuronal functions.

## Methods

**Animals**. All experiments were conducted in accordance with National and European (2010/63/EU) laws for the use of animals in research and were approved by the local ethical committee (Landesuntersuchungsamt Rheinland-Pfalz, 23 177-07/G 14-1-080 and G 20-1-006). Offspring from timed-pregnant C57BL/6NRj wild-type mice (Janvier Labs) were used for all experiments.

**Primary cortical cultures**. Experiments were performed in primary cortical neurons cultured from newborn (postnatal day 0) mice. After decapitation, brains were transferred to ice-cold $Ca^{2+}$-and $Mg^{2+}$-free HBSS (Gibco, Invitrogen, Carlsbad, CA, USA) supplemented with penicillin and streptomycin (50 units/ml), sodium pyruvate (11 mg/ml), glucose (0.1%), and HEPES (10 mM). Cortical cells were dissociated via trypsin incubation for 20 min at 37 °C and DNAse digestion at room temperature (RT). After blocking trypsinization by washing steps with HBSS, Minimal Essential Medium (MEM, Gibco) was supplemented with 10% horse serum and 0.6% glucose. Next, cells were mechanically dissociated via repetitive pipetting through fire-polished glass pipettes with declining diameter. Cells were counted after trypan blue staining and seeded on IBIDI dishes with an initial plating density of approximately 1000 cells per mm² on coverslips (μ-Dish 35 mm Grid-500 Glass Bottom, μ-Slide 8 well Grid-500, Ibidi, Germany). Coverslips and IBIDI-dishes were coated with poly-ornithine prior to plating of cells. After 45 min, the medium was exchanged for medium consisting of Neurobasal medium (Gibco) supplemented with 2% B27 (Gibco) and 1 mM L-glutamine. Cells were cultivated at 37 °C in humidified carbogen (95% air; 5% $CO_2$) for up to 9 days. After two days in vitro (DIV 2) 5 μM Cytarabine (AraC) was added to the medium to inhibit glial cell proliferation. Half of the medium was replaced by BrainPhys™ Neuronal Medium (supplemented with SM1 supplement; Stem cell technologies, Vancouver, Canada) on DIV7 for dishes that were imaged on DIV9.

Primary neurons were transduced with recombinant adeno-associated virus (AAV) carrying pAAV-hSyn-H2B::mCherry after one day in culture. Imaging experiments were performed on DIV8 or DIV9. Penicillin and streptomycin (5 units/ml) were applied prior to the imaging session to avoid bacterial contamination.

**Mouse brain sample preparation**. For tissue SMLM imaging, mice were transcardially perfused with PBS supplemented with heparin followed by 4% paraformaldehyde (PFA) fixation. Brains were removed and post-fixed overnight in 4% PFA at 4 °C. Afterwards, left and right cortices were separated and lipids were removed via incubation in Cubic-L (TCI, Deutschland), and the tissue was treated with the Ambion RNase Cocktail (Invitrogen, RNase A = 0.5U/ml, RNase T1 = 20U/ml) at 4 °C overnight. Cortices were stained with a rabbit polyclonal anti-cleaved Caspase 3 (aCasp3) antibody 1:2500 (Cell Signaling Technology, Inc., San Diego, CA, USA) for 2.5 days, followed by Alexa Fluor 647 AffiniPure donkey anti-rabbit polyclonal antibody 1:1300 (Jackson Immuno Research) for 2 days. The intact cortices were then prepared for cryosections with 10% sucrose in PBS for 0.5 days and 30% sucrose in PBS overnight, and finally embedded in clear Tissue-Tek O.C.T. compound (Sakura), before being sliced into 14-μm thick cryosections. Prior to imaging, cryosections were stained with Sytox Orange Nucleic Acid Stain (Invitrogen, 0.1 μM).

**Confocal time lapse acquisition**. High resolution confocal images were acquired with a Visiscope 5-Elements spinning disk confocal system (Visitron Systems, Germany), and a 60× water immersion objective (CFI Plan Apo VC 60XWI, 1.2 NA). The system was equipped with Yokogawa CSU-W1 scan head, a Prime BSI sCMOS camera (2048 × 2048 pixels, 6.5 μm pixel size, Photometrics), and an incubator (Bold Line universal stage top incubator, okolab, Italy). Lateral resolution was 163 nm and axial resolution was 490 nm in a field of view of 228 × 228 μm. The laser lines of 488 nm and 561 nm were used for the fluorescence excitation and the fluorescence emission was filtered using filters 525/30 nm bandpass (Chroma) and 570 nm longpass (Chroma) for NucView and H2B::mCherry, respectively. The temperature in the incubator was kept constant at 37 °C and the $CO_2$ level at 5%. Images were acquired every 10 min for a total duration of 7 h. 7 z-planes with a distance of 2 μm were acquired per time point. For live-time detection of caspase activity in all experiments, the green fluorescent caspase-3/7 substrate NucView (Biotium, 1.6 μM) was added. Thus, apoptotic events could be detected based on a transient elevation of the NucView signal.

Neurons were fixed with 2% PFA for 5 min and 4% PFA for 25 min immediately after the live cell acquisition. For a subset of experiments, fixed

neurons were subsequently imaged with the SMLM super resolution technique as described below.

In general, neurons were kept for 30 min untreated for a baseline acquisition during confocal live cell imaging. Afterwards, pharmacological treatments were performed as follows and neurons were imaged for an additional 39 time points over the course of 6.5 h.

**Pharmacology**. Upon acquisition of baseline, neurons were either treated with 1.5 μM staurosporine (Sigma-Aldrich) to induce apoptosis or kept under control conditions. In a sub-set of experiments, neurons were additionally treated with caspase inhibitor (100 μM, Z-DEVD-FMK, R&D Systems), and the myosin blocker BDM (2,3-Butanedione menoxime, Sigma-Aldrich, 20 mM) in the presence of staurosporine.

For experiments in which confocal imaging preceded SMLM acquisition, H2B::mCherry-transduced primary cortical neurons were either imaged as control (1.6 μM NucView and imaging for 1 h) or treated with staurosporine. In this case, 1.0–1.5 μM staurosporine was applied to the dishes 0 to 4 h before the acquisition started, then 1.6 μM NucView was added and the neurons were imaged for up to 6 h. All cells were directly fixed with 4% PFA afterwards.

**Immunocytochemistry and confocal imaging of fixed neurons**. For a subset of experiments, neurons were fixed under control condition, or after 2 h or 6 h upon induction of apoptosis by staurosporine with a stepwise PFA fixation (5 min 2% then 20 min 4% PFA) followed by a subsequent wash with PBS. Fixed neurons were then stained with (ms) monoclonal anti-trimethyl Histone H3 (Lys27) (1:200; Sigma Aldrich), rabbit polyclonal anti-non-muscle myosin IIA (1:200; ab75590 Abcam) or rabbit polyclonal anti-non-muscle myosin IC (1:200; Thermo Fisher Scientific) after blockade of unspecific binding of antibodies with 7% normal donkey serum and 0.3% triton diluted in PBS for 2 hr at RT and followed by Alexa Fluor 647 AffiniPure donkey anti-mouse polyclonal antibody or Alexa Fluor 647 AffiniPure donkey anti-rabbit polyclonal antibody (1:500; Jackson Immuno Research) and DAPI (300 nM, Thermo Fisher Scientific).

Confocal imaging was performed at a Visiscope 5-Elements spinning disk confocal system with a 100× oil immersion objective (CFI SR HP Apochromat TIRF, NA 1.49). Co-localization of nuclear trimethyl Histone H3 signal of single neurons was analyzed by the toolbox for subcellular co-localization analysis under Image J (JACoP[80]). For quantification of nuclear myosin signal intensities at different experimental conditions, nuclear signal intensities were analyzed in Image J based on average z-projections of confocal stacks encompassing the entire thickness of each neuron and by delineation of nuclear boundaries based on DAPI signal.

**Caspase Glo and cell viability (Alamar Blue) assay**. Caspase activity was measured by a luminescent Caspase-Glo®® 3/7 Assay (Promega, Madison, WI, USA) in DIV8 neurons under baseline conditions, as well as 3 h and 6 h after start of pharmacological treatments. This homogeneous, luminescent assay provided a luminogenic caspase-3/7 substrate, which contained the tetrapeptide sequence DEVD, in a reagent optimized for caspase activity, luciferase activity, and cell lysis. The protocol was performed according to the manufacturer's guidelines. In short, equal amounts of Caspase-Glo® 3/7 reagent and PBS were added to the cells upon removal of the medium. Buffer and cells were mixed and incubated on the samples for 30–45 min. Luminescence was measured with a Tecan infinite M1000 plate reader (Tecan, Maennedorf, Switzerland) and normalized to untreated control for statistical analysis.

Viability of primary neuronal cultures was assessed by the redox indicator of cell viability Alamar Blue HS Assay (Life Technologies, Schwerte, Germany). Medium was substituted with Alamar Blue HS reagent (1:10). Neurons with Alamar Blue were incubated at 37 °C in humidified 95% air and 5% $CO_2$ for 45–60 min. Fluorescence was analyzed with an automated Tecan plate reader (Tecan GmbH, Crailsheim, Germany) with excitation wavelength of 540 nm and emission of 595 nm. Background was subtracted and all values were normalized to averaged H2B::mCherry-transduced control.

**SMLM**. For SRM experiments, a custom-built inverted microscope setup was used[81], without the patterned illumination. In short, the microscope was equipped with a 488 nm, a 561 nm and a 647 nm laser. Lasers were combined using dichroic mirrors and expanded by a pair of lenses and focused to the back focal plane of the microscope objective (HCX PL APO 100×/NA 1.47 OIL, Leica, Germany). A quadband dichroic mirror reflected the excitation light to the sample and transmitted fluorescence light to the detector. Emission filters were used to block unwanted stray light and to separate fluorescence in multicolor experiments. A tube lens then focused the fluorescence light onto a sCMOS camera, with an effective pixel size of 65 nm. The 561 nm laser was used at 1.3–2.8 kW/cm², 50 ms exposure time, and 30,000 frames were acquired for primary cortical neurons, while 60,000 frames were acquired for the mouse cortical cryosections. On the day of SMLM acquisition, 143 mM β-Mercaptoethanol (Sigma-Aldrich) in PBS was applied to the fixed primary

cortical culture dishes as a blinking buffer for the H2B::mCherry signal, and mouse brain cryosections were mounted in a blinking buffer for fluctuation binding-activated localization microscopy[55] (fBALM), optimized for the Sytox Orange nucleic acid stain[81].

**Analysis of confocal imaging data**
*Pre-processing.* The acquired 7 z-planes of each FOV were z-projected onto one final image, aligned with hyperstackreg[82], and cut out in equal ROIs in ImageJ[83]. Individual ROIs were run through a Matlab analysis pipeline adapted from Irianto et al.[27], with a pixel factor of 2, measuring nuclear size (area), edge count, chromatin compaction parameter (CCP), and caspase activation (NucView signal intensity).

*Calculation of apoptotic neurons.* Identification of apoptotic neurons and percentage per culture was calculated based on the transient elevation of the NucView signal. A threshold of 14-fold increase in the normalized NucView signal was selected and neurons that surpassed it were counted as apoptotic.

*Chromatin compaction parameter (CCP).* For analysis of the degree of chromatin compaction, the CCP analysis pipeline described by Irianto et al.[27] was adapted to confocal spinning disk data. The CCP analysis is based on a Sobel edge detection and assumes that the chromatin compaction leads to an increase of distinct spaces within the nucleus. This increase can then be quantified by measuring sharp intensity transitions (edges) within the nucleus. The CCP is defined as the number of pixels that contain edges divided by the cross-section area of the respective nucleus. The algorithm was modified to analyze data with the different noise properties of the sCMOS camera used. Essentially, a thresholding step based on Otsu's method was implemented to segment cell nuclei before calculating CCP values. For each frame of the time lapse, a mask was created to include only the pixels containing the cell of interest in the red, H2B::mCherry channel. ROIs that contained more than one nucleus covered by the binary mask were excluded from the analysis. From each mask, the area and number of edges were counted, and the chromatin compaction parameter (edges divided by area) was calculated. Additionally, the mean intensity of the caspase 3 activation (NucView channel) inside the mask was calculated for each frame.

For the comparison of kinetics of CCP, nuclear size and edge count over the course of different cell fates (control, apoptosis and necrosis, Fig. 1e–h), signals of nuclei of different cells from different experiments were aligned to the time point of first clear change in nuclear size (relative t = 0 h). Confocal time-lapse images and quantitative analysis of a single representative neuron confirm that the increase in chromatin compaction precedes the decrease in nuclear size and caspase activation on single neuron level (Suppl. Figure 5).

*Cell fate classification of neuronal nuclei.* Based on the mean intensity of NucView signal, nuclear size and CCP during confocal live cell imaging, neurons were assigned in a step-wise process to either of the following four categories: apoptotic, necrotic, no change, and granulating. All values were normalized to the baseline (average of the 3 baseline recordings).

Step 1: Neurons were classified to be "dying or alive" according to the presence or absence of a relevant change in the normalized NucView signal per cell, calculated by the finite difference method. Neurons were classified as "surviving" if the normalized gradient between consecutive time points during the entire imaging session was below threshold of 0.035. Neurons were labeled as "dying" if the normalized gradient of consecutive timepoints at the end of recording was above 0.8, because any death of neurons is followed by an increase in NucView signal. No label was given if there was no clear indication for either category.

Step 2: Neurons were categorized as "apoptotic" if the normalized CCP value at the last time point of acquisition was smaller than 1.1 fold of CCP at the first time point, a strong indicator that no granules were present at the last time point.

Step 3: Neurons were classified as "non-apoptotic, necrotic-like" if the normalized nuclear size value at the last time point decreased to least 75% of the first time point, indicating that the nucleus shrank.

Step 4: Neurons were categorized as "granulating" if the normalized CCP value at the last time point increased by at least 50% of normalized CCP at the first time point, and as "control" if the difference in normalized CCP value was less than 50%. Within the time course of the experiments, an elevated CCP indicated high chromatin compaction (referred to as granulation) and the possibility of a later apoptotic or necrotic death. Unaffected CCP values indicated no change in chromatin morphology.

Step 5:

a. Apoptotic nuclei: Neurons were defined as "apoptotic" only if neuronal nuclei were labeled as "dying" in step 1 and "apoptotic" in step 2.

b. Necrotic nuclei: Neurons were considered "necrotic" if neuronal nuclei were labeled as "necrotic" in step 3, not labeled as "surviving" in step 1, and if they had not been defined as "apoptotic" in (a).

c. Granulating nuclei: Neuronal nuclei were defined as "granulating" if they were classified as "granulating" in step 4 and they had not been defined prior to be "apoptotic" (a) or "necrotic" (b).

 d. "No change" nuclei: Neurons were defined as "no change" if they were classified as "control" in step 4 and they were not defined as "apoptotic" (a), "necrotic" (b), nor "granulating" (c), previously.

*Classification of apoptotic stages before SRM.* Based on confocal time lapse images preceding the super resolution acquisition, the neuronal nuclei were manually classified into five apoptotic stages, with stage 1 being control condition. These stages were defined based on the literature[8] and our own observations from the confocal live cell imaging data. Machine learning algorithms for pixel and object classification in ilastik[50] were used to test the by-eye classification through automatic classification of neuronal nuclei into the five stages of apoptosis. Here, a set of 11 neuronal nuclei were used exclusively for training and not included further in the analysis. Only nuclei that were classified into the same stage by both the human eye and machine learning based on confocal images were included into further analysis with SMLM.

### Analysis of super-resolution imaging data

*Pre-processing.* Image reconstruction of SMLM acquisitions was performed using the ThunderSTORM plugin[84] in ImageJ. The threshold for the standard deviation of the wavelet filter for reconstruction was set at 1.0 for fixed primary neurons, and 0.7 for cryosections. To avoid multiple detections of the same molecules, the merge function of ThunderSTORM was used, with maximum distance set at 20 nm and maximum off frames set at 1; the maximum number of frames per molecule was unlimited. False localization signals were prevented by filtering out localizations with Full-Width-at-Half Maxima (FWHMs) that were smaller than 65 nm or bigger than 225 nm, values that would not be compatible with theoretical PSF predictions. Blinking events outside the nucleus were cut out with a filtering on the x and y coordinates when necessary.

*Voronoi analysis.* The localizations from ThunderSTORM were handed over to a custom python analysis script in *csv* format. To make the individual reconstructed nuclei comparable, we transformed the cartesian coordinates into polar coordinates, using the center of mass of all localizations of each individual nucleus as the origin of the polar coordinate system. To calculate the density of the localizations, we used the Voronoi tessellation using the Scipy[85] class *scipy.spatial.Voronoi* for the localizations of each individual nucleus. Gauss's area formula was used to calculate the area of each Voronoi cell from its vertices and its inverse as the Voronoi density for each localization. The density values are confounded by the number of localizations of the particular nucleus. Thus, for normalization the Voronoi densities were divided by the number of localizations of the particular nucleus. Finally, a log transformation was applied to support the visualization. Voronoi density histograms were plotted using the *seaborn.histplot* function from the seaborn package[86]. The Voronoi density axis always contains the log-transformed localization count normalized Voronoi density data. To account for variations in the number of localizations in different stages, the probability density function was calculated and normalized such as the area under the histogram is 1. For a two-dimensional visualization, histograms were plotted using the seaborn.histplot function with the same count normalization as in the one dimensional case, but including the radius on the y-axis and the probabilities as the color-code of the graph. Limits for the radius axis: 0 nm to 8000 nm), the log-normalized density axis: $-10 \log(nm^{-2})$ to $-5 \log(nm^{-2})$ and for the color bar of probability density function: $1 \times 10^{-5}$ to $1 \times 10^{-3}$.

### Statistics and reproducibility.

All confocal data were analyzed with Prism (GraphPad Software) and are expressed as mean ± SEM. Error bars on graphs show the SEM. *p* values of less than 0.05 were considered to be statistically significant, whereas *p* values greater than 0.05 were considered non-significant (n.s.). Experimental groups consisted of at least three biological replicates gathered from minimum three independent experiments. Statistical analysis of two experimental groups was performed with the parametric two-tailed Student's t-test; when more than 2 groups were compared, we performed a one-way or two-way analysis of variance (ANOVA), a mixed-effects analysis, or Kruskal-Wallis test when applicable, and differences between groups were analyzed by a subsequent Tukey's, Dunn´s or Sidaks test for multiple comparisons.

 SRM data were analyzed with Python. For log-transformed normalized Voronoi densities of apoptotic stages mean and SD are reported. Results for groups of SRM data were statistically compared with non-parametric pairwise comparisons Wicoxon rank sums test. Statistical analysis of Voronoi density distributions of cultured neurons with Wilcoxon rank sums resulted in extremely low *p*-values even lower than the floating point precision in python $(2.23 \times 10^{-308})$ because the large number of blinking events (total localizations stage 1 $3.51 \times 10^5 \pm 4.43 \times 10^4$; stage 2 $2.12 \times 10^5 \pm 1.57 \times 10^4$; stage 3 $1.76 \times 10^5 \pm 5.28 \times 10^4$; stage 4 $3.62 \times 10^5 \pm 1.82 \times 10^5$; stage 5 $2.05 \times 10^5 \pm 7.04 \times 10^4$). Also aCasp3-negative and aCasp3-positive neurons in vivo displayed a large number of blinking events $(3.45 \times 10^5 \pm 1.34 \times 10^4$ localizations, and $2.16 \times 10^5 \pm 1.14 \times 10^5$ localizations) used for statistical analysis of Voronoi density distributions using Wilcoxon rank sums yielded extremely low p-values. Therefore, we omitted the interpretation of statistical results for the comparison of Voronoi density distributions between groups.

**Reporting summary.** Further information on experimental design is available in the Nature Research Reporting Summary linked to this paper.

## Code availability

The code used for analysis of the super resolution data is also available in the ZENODO repository, https://doi.org/10.5281/zenodo.5603044.

## Data availability

The datasets generated during and analyzed during the current study are available in the ZENODO repository, https://doi.org/10.5281/zenodo.6673787.

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

## Acknowledgements

We kindly thank Simone Dahms-Prätorius and Beate Krumm for the excellent technical assistance and Fridolin Kielisch for the discussion on statistical analysis. Support by the IMB Microscopy Core Facility is fully acknowledged. We gratefully acknowledge Shih-Ya Chen's technical help regarding SMLM and helpful discussions. Acknowledgement to Udo Birk, Jan Neumann, and Amine Gourram for contributions to SMLM analysis, and to Ole Kröger and Filip Sadlo for contributions to Voronoi analysis. This work was supported by a grant of the Deutsche Forschungsgemeinschaft to H.J.L. and A.S. (CRC1080, project A01). The Spinning Disk Confocal System (VisiScope, 5-Elements, IMB Microscopy Core Facility) was supported by the Deutsche Forschungsgemeinschaft (INST 247/912-1FUGG) as well. R.R. is a graduate student in the International PhD Program at the Institute of Molecular Biology in Mainz and E.N. is a graduate student of the TransMed PhD Program of the University Medical Center in Mainz. The work of C.C. and M.G. was supported by the Boehringer Ingelheim Foundation.

## Author contributions

Conceptualization, R.R., H.J.L., and A.S.; Methodology, R.R., A.S., E.N., C.C., M.G., and S.R.; Formal analysis, R.R., A.S., E.N., N.P., and M.G.; Investigation, R.R. and A.S.; Writing—original draft, R.R. and A.S.; Writing—review & editing, H.J.L., A.S., and C.C.; Visualization, R.R., N.P., S.R., and A.S.; Supervision, A.S., H.J.L., and C.C.; Funding acquisition, A.S., H.J.L., and C.C.

## Funding

## Competing interests

The authors declare no competing interests.
