## [Peer Review File · Communications Biology]

Reviewers' comments:

Reviewer #1 (Remarks to the Author):

In this paper, the authors observed changes in the nuclear structure of neurons during apoptosis. Time lapse imaging of neuronal cultures in vitro revealed that the nuclear structure changes in apoptotic cells even before the activation of Caspase. This change was not inhibited by the Caspase inhibitor, indicating that it is an event independent of the execution of apoptosis by Caspase. They also classified the structural changes and analyzed the nuclear structure in an in vivo tissue section.

Although the authors observe an interesting phenomenon, namely, changes in nuclear structure, which may lead to the mechanism of neuronal apoptosis, the method of analysis and the use of language are vague and insufficient to support the authors' claims. The following points need to be addressed.

Major points

1. Which does "chromatin condensation/compaction" in this paper mean decrease of chromatin accessibility as examined by DNase-seq and ATAC-seq or the accumulation of chromatin fibers? The authors should show what kind of chromatin structural changes they are observing in this paper, for example, by using ATAC-seq (Chen, Nat. Method, 2016) or FRET between histone proteins.
2. How are "edge" and "granulation" defined and what is the difference and relationship between them? Also, what is the difference and relationship between them and the parameter in Volonoi analysis? The definition of each word and its biological significance are unclear. In addition, the authors need to show a scheme diagram of what Volonoi density represents.
3. In relation to the above, CCP (chromatin condensation parameter) is misleading. It should be written as "edge density".
4. As mentioned by the authors in the Discussion, BDM experiments do not allow to conclude that nuclear myosin contributes to nuclear structural changes, as mentioned by the authors in the abstract, such as "concurrent interference with the dynamics of chromatin organization by blocking actomyosin activity prevented apoptosis but resulted in necrotic-like cell death instead." and "this early condensation is essential for but not part of the execution of apoptotic cell death in developing neurons." To conclude this, the following experiment should be performed.
 - a. Immunostaining of nuclear myosin during apoptosis to show how myosin is related to changes in nuclear structure.
 - b. Is the function of nuclear myosin indeed inhibited by BDM treatment?
 - c. Do the authors observe the similar results to Figure 2 when nuclear myosin is specifically inhibited?
5. In Figure 1e-h, "nuclear shrinkage in apoptotic cells" and "swelling in necrotic cells" are defined as time = 0. Are they comparable events? The authors should explain why they defined time = 0 for these.
6. In Figure 1f, when does the edge count start to increase? Was there ever a time when the edge count of these cells was the same as that of the control cells?
7. In Figure 4a, all stages should be examined in a same cell. In addition, the stage should be defined using quantitative parameters obtained from multiple cells.
8. What is the novel point of in vivo analysis? Based on the in vitro analysis, the authors should examine the changes in nuclear structure of the cells undergoing apoptosis before Caspase activation.

Minor points

9. Expression of H2B-mCherry is toxic, so it should be verified by other methods as well.
10. Between which samples does the * in the Figures show the significant difference? What do (condition) or (treatment) mean?
11. Figure 4b and Figure 6c are meaningless because they quantify the signals in arbitrary and very local area.

12. In Discussion, there is no evidence that transcriptional regulation via chromatin structure is important for the significance of the observed changes in nuclear structure prior to Caspase activation. Other possibilities should be discussed.

13. This reviewer do not understand the meaning of the last paragraph of the Discussion. The authors should discuss how this study can contribute to the understanding of "brain function, dysfunction, and development".

Reviewer #2 (Remarks to the Author):

The authors are interested in the underexplored phenomenon of nuclear and chromatin condensation that occurs during cellular apoptosis. To investigate this, they employ confocal microscopy of neurons expressing an H2B-mCherry fusion, along with super-resolution imaging of fixed neurons. They investigate chromatin condensation and nuclear morphology in these cells upon treatment with staurosporine to induce apoptosis, with or without co-treatment with either an inhibitor of caspase activation or actomyosin activity. Via these techniques the authors divide the process of apoptotic nuclear condensation into 5 stages. Most notably, the earlier stages occur prior to changes in nuclear morphology (e.g. shrinking), and the earliest prior to caspase activation. Caspase inhibition does not inhibit this earliest stage of chromatin condensation, while actomyosin inhibition – here used to prevent the movement of “nuclear bodies” – prevents apoptosis and results in a necrosis-like phenotype.

This study addresses a genuinely underexplored topic in chromatin condensation that accompanies apoptotic cell death. To give some perspective on how glacial progress on this topic has been, a protein termed Acinus was first characterized as playing an important role in apoptotic chromatin condensation in 1999, but the number of research articles dealing with its function in this regard can be counted on the fingers of one hand, and I would argue we still know nothing about what this factor does. So the research topic and the approaches being used here are overdue and most welcome. To a certain extent, the findings reported here raise more questions than they answer, but as an initial foray into this subject I still think this study serves well. There are issues with clarity and possibly logic that I think need to be addressed, however (see specific comments, below).

Comments:

1. In Fig. 2, I’m not sure I see how the caspase inhibitor results in behavior similar to “early granulation observed in staurosporine-only treated cultures.” The graphs show that the similarity runs throughout the time course of the experiment. Is something not clear here? Is the time course shown in Fig. 2 intended to be limited to an “early” phase of apoptosis, and if the authors showed later time points a difference would emerge?

2. It’s a bit of a stretch to go from this experiment to the conclusion that the early chromatin compaction is necessary for apoptosis. This shows that actomyosin is necessary, and proximally perhaps that movement of “nuclear bodies” is also necessary. Compaction may be the operative principle, but a lot of other possibilities, starting with transcriptional regulation, are still likely (to take one possibility: does the initial stage of compaction simply reflect a global loss of transcription?). The authors do include a disclaimer to this effect, however. Still, it isn’t clear what this experiment says about the timing involved – that is, doesn’t application of BDM inhibit nuclear reorganization both before and after caspase activation? How do the authors then conclude that the reorganization requirement implied by the BDM experiment applies to the caspase-independent component? At the very least, this point needs clarification.

3. Similarly, I do not understand how the authors conclude that the initial compaction is somehow “not part of the final execution of the apoptotic process per se.” How can they make this distinction? What

does this even mean?

4. The transition from compaction stages 4 to 5 (Fig. 4) is confusing. What is happening in stage 4? Does this represent discontinuous staining of chromatin within membrane-bounded nuclei or a fragmentation of the nuclei? How do discrete nuclei then appear again at stage 5? Additional clarification is needed here.

Reviewer #3 (Remarks to the Author):

Communications Biology/Nature
COMMSBIO-21-2691

The authors described dynamic changes in chromatin organization that lead to the chromatin cluster formations that appear early in developing neurons with a prospective apoptotic fate, although they are not properly part of the final execution of the apoptotic process. Such a chromatin reorganization precedes caspase-3 activation and is distinct from the continuous condensation of the nuclei. The manuscript is a novel contribution of fundamental importance to the understanding of the apoptosis phenomenon. Overall, it is a major advance in this field. The article is nicely written and technically accurate. It was well planned and executed. Sophisticated chromatin conformation capture techniques were used for the evaluation of the preparations. The results are well described and discussed with adequate literature survey. Although I have no expertise in the Voronoi analysis and respective statistics I can understand the importance of the authors' contribution.

Minor:

- Many sequential words and symbols in the text appear frequently not separated by space. Ex.: 163nm, was490, etc.

- Materials and Methods

line 632 – Please write AraC in full the first time its mentioned (cytarabine).

line 636 – Please write AAV in full.

line 718 – Remove “chromatin condensation parameter”. CCP has been defined in the preceding line.

Standardize: minutes or min; hours or h?

- Results

line 114 – “nucleus” swelling

Fig. 1. For easier reader identification of relative time in e and f, please insert values in the X line

- References

line 918 – abbreviate the name of the journal

Reviewer #1 (Remarks to the Author):

In this paper, the authors observed changes in the nuclear structure of neurons during apoptosis. Time lapse imaging of neuronal cultures in vitro revealed that the nuclear structure changes in apoptotic cells even before the activation of Caspase. This change was not inhibited by the Caspase inhibitor, indicating that it is an event independent of the execution of apoptosis by Caspase. They also classified the structural changes and analyzed the nuclear structure in an in vivo tissue section.

Although the authors observe an interesting phenomenon, namely, changes in nuclear structure, which may lead to the mechanism of neuronal apoptosis, the method of analysis and the use of language are vague and insufficient to support the authors' claims. The following points need to be addressed.

Major points

1. Which does "chromatin condensation/compaction" in this paper mean decrease of chromatin accessibility as examined by DNase-seq and ATAC-seq or the accumulation of chromatin fibers? The authors should show what kind of chromatin structural changes they are observing in this paper, for example, by using ATACsee (Chen, Nat. Method, 2016) or FRET between histone proteins.

We agree that the correlation of spatial chromatin organization, molecular signatures and most optimally also cellular state or neuronal functions would be of highest interest, in the context of apoptosis but also under many different physiological and pathophysiological conditions. While the proposed experiments to address spatial organization of chromatin structure on a genome-wide scale by ATAC-seq or chromatin conformation by Hi-C would be generally interesting, a necessary longitudinal analysis of a dynamic process on the single neuron level is to our knowledge not possible. Therefore, we have initially decided for the longitudinal description of structural changes of chromatin by high resolution microscopy. Changes in chromatin compaction of a given single neuron can be reliably captured through an increase in H2B::mCherry fluorescence signals in confocal live cell experiments in a longitudinal manner and also under pharmacological manipulations or increased density of blinking events in a SRM (Super-Resolution Microscopy) approach after fixation. The quantification of chromatin compaction is possible since the condensation of chromatin is associated with an increase of spaces (i.e. histones captured by H2B::mCherry signal density or number of DNA bound Sytox Orange molecule positions). Detection of these spaces through a Sobel edge detection algorithm as density of edges within the nucleus, normalization to its cross-section area defined as the chromatin condensation parameter thus gives a reliable measure of the level of chromatin condensation ²⁷. For confirmation, immunocytochemical co-stainings with the heterochromatin marker trimethyl-Histone H3 (Lys27) ³¹ confirmed high levels of co-localizations under control and apoptotic conditions (new Suppl. Fig. 2, Pearson's Coefficient: control 0.93 ± 0.004 apoptosis 0.86 ± 0.02 , n=5/7 cells). For the quantification of chromatin compaction based on super resolution data Voronoi tessellation of localizations of individual blinking events (i.e., location of chromatin-derived histone signals or DNA bound Sytox Orange molecule positions) within the neuronal nuclei were calculated based on the SMLM reconstructions (Fig. 5a). This analysis thus allows also the systematic comparison of the density of single H2B::mCherry molecules within nuclei and thus chromatin compaction although not in a longitudinal manner but of identified neurons before and during the execution of apoptosis and under control conditions. For both, SRM and confocal analysis of chromatin-derived histone signals or DNA bound Sytox Orange molecule

positions signals were normalized throughout longitudinal recordings. Further, the area under the curve of distributions of log-normalized Voronoi densities was normalized and the probability density function analyzed to correct for different number of localizations cross cells thus an accumulation of chromatin fibers can be excluded.

In general, only an appropriate chromatin density provides the steric conditions for an appropriate mobility of these macromolecules and macromolecular aggregates and thus determines the accessibility of specific intranuclear chromatin domains for individual macromolecules and macromolecular aggregates. The initiation of transcription, for example, requires the interaction between transcription factors and target DNA sequences. The same is true for other essential nuclear functions, e.g., for replication or repair. According to theoretical models (e.g. ANC-INC model ^{14, 16}), genomic targets hidden in the compact interior (INC, Inactive Nuclear Compartment, high DNA density) of chromatin domains or still larger chromatin ensembles may be accessible for individual transcription factors, but inaccessible for functional protein/RNA ensembles. The differential accessibility hypothesis of chromatin ensembles located in the ANC (Active Nuclear Compartment, low DNA density) and in the INC suggest an additional level of spatial fine-tuning of transcriptional control, as well as of other nuclear processes, such as chromatin remodeling, replication, and repair ¹⁶. Thus, we conclude that local intranuclear DNA and thus chromatin density is directly correlated with the probability of transcription of respective gene and hence of gene expression. On a larger scale a strong chromatin compaction is likely associated with an overall transcriptional downregulation.

We have included this information in the introduction and included new data confirming the co-localization of H2B::mCherry signal with the heterochromatin marker trimethyl-Histone H3 (Lys27) to the results section and as a new supplemental figure 2 and have added a more refined outlook to the discussion of the present version of this study.

14. Cremer, T. *et al.* The 4D nucleome: Evidence for a dynamic nuclear landscape based on co-aligned active and inactive nuclear compartments. *FEBS Lett.* **589**, 2931–2943 (2015).
16. Cremer, T. *et al.* The Interchromatin Compartment Participates in the Structural and Functional Organization of the Cell Nucleus. *BioEssays* **42**, e1900132 (2020).
27. Irianto, J., Lee, D. A. & Knight, M. M. Quantification of chromatin condensation level by image processing. *Med. Eng. Phys.* **36**, 412–417 (2014).
31. Margueron, R. & Reinberg, D. The Polycomb complex PRC2 and its mark in life. *Nature* **469**, 343–349 (2011).

2. How are "edge" and "granulation" defined and what is the difference and relationship between them? Also, what is the difference and relationship between them and the parameter in Volonoi analysis? The definition of each word and its biological significance are unclear. In addition, the authors need to show a scheme diagram of what Volonoi density represents.

We apologize if the terminology in the previous version of the manuscript lead to misunderstandings or was not clear enough. **An increase in chromatin condensation can be visually described as chromatin granulation.** To quantify the level of compaction of chromatin of single neurons before and during cell death based on high resolution confocal imaging data, we performed a Sobel edge detection algorithm on the H2B::mCherry signal to longitudinally **assess the number of H2B::mCherry fluorescent signal within the nucleus (i.e. edges)**. An increase in chromatin compaction is therefore captured as an increase H2B::mCherry signal i.e. histone density. As discussed previously an increase in chromatin compaction can be interpreted as a decrease in DNA

accessibility. For the quantification of chromatin compaction based on super resolution data, Voronoi tessellation of the localizations of individual blinking events (i.e., location of chromatin-derived histone signals or DNA bound Sytox Orange molecule positions) within the neuronal nuclei were calculated based on the SMLM reconstructions (Fig. 5a). This analysis allows the systematic comparison of the density of H2B::mCherry molecules within nuclei before and during the execution of apoptosis and under control conditions. To account for different numbers of localizations and cells for each stage, the area under the curve had to be normalized to 1 and the probability density function was plotted. **An increase in chromatin condensation that is characterized by increasing edge counts and CCP based on confocal imaging would be represented by a right-shift of the peak of probability density function** due to an increase in density of individual H2B::mCherry or DNA bound Sytox Orange molecule positions localizations. Again, an increase in chromatin compaction results in denser and bigger cluster sizes (i.e. decrease in Voronoi tessellation area sizes and thus a right-shift in densities of individual H2B::mCherry localizations) is associated with a decrease in DNA accessibility.

We have now included a schematic illustration of how Voronoi tessellation is computed and used for calculation of Voronoi densities based on single molecule localizations in the **revised Fig. 6a** and have **changed figure legends and results sections (line 376ff) to facilitate the understanding and interpretation of the results of the Voronoi-based analysis of the super resolution data.** Also, we included a short description on how these results relate to the quantitative results on detected edges and CCP and the granular appearance of the nuclei from the confocal imaging.

3. In relation to the above, CCP (chromatin condensation parameter) is misleading. It should be written as "edge density".

CCP (chromatin condensation parameter) was previously introduced by Irianto and colleagues as a direct measure for the level of chromatin compaction²⁷. Although we surely agree that this measure can also be described as "edge density" we would prefer to keep the original and thus uniform nomenclature. **To improve clarity, we have included a more detailed description as follows to the results section:**

Results 141ff: *To quantify the level of compaction of chromatin of single neurons before and during cell death, we first performed a Sobel edge detection algorithm on the H2B::mCherry signal to longitudinally assess the number of chromatin-associated fluorescence signals as edges within the nucleus (i.e. edge count). Then, we determined the density of edges within the nucleus by normalization of the detected edges to the cross-section area (i.e. edge count/nuclear size; see also Fig. 1d and Methods), and thereby calculated a direct measure for the level of chromatin compaction which was previously defined as chromatin condensation parameter (CCP)²⁷.*

27. Irianto, J., Lee, D. A. & Knight, M. M. Quantification of chromatin condensation level by image processing. *Med. Eng. Phys.* **36**, 412–417 (2014).

4. As mentioned by the authors in the Discussion, BDM experiments do not allow to conclude that nuclear myosin contributes to nuclear structural changes, as mentioned by the authors in the **abstract**, such as "concurrent interference with the dynamics of chromatin organization by blocking actomyosin activity prevented apoptosis but resulted in necrotic-like cell death instead." and "this early condensation is essential for but not part of the execution of apoptotic cell death in developing neurons." To conclude this, the following experiment should be performed.

The abstract and conclusions in the respective results sections haven been carefully revised.

a. Immunostaining of nuclear myosin during apoptosis to show how myosin is related to changes in nuclear structure.

We thank the reviewer for this helpful suggestion for experiments addressing the subcellular localization of myosin and chromatin structure during the apoptotic process. Besides the conventional myosin II, at least 7 of the known unconventional myosins are expressed in the nucleus albeit with largely unknown functions (reviewed recently by Maly and Hoffmann 2020, Cook et al. 2020)^{45,46}.

Specifically, we performed immunohistochemical co-stainings with the conventional myosin IC and the unconventional myosin IIA of control neurons and neurons in the early vs. late phase of apoptosis. The results, which we now included as new Fig. 4 and Supplemental Fig. 3, confirm the expression of myosin IIA and IC in the nucleus of primary cortical neurons. Within the nucleus, myosin IIA and myosin IC signal were not correlated with the H2B::mCherry signal but rather complementary expressed (See Fig. 4a and Suppl Fig 3a). While we did not observe significant changes in the expression of the conventional myosin IIA during apoptosis (Suppl Fig. 3a-c), the nuclear myosin IC signal intensity significantly decreased with the progression of the apoptotic process (Fig. 4a-c). Although the reduction of nuclear myosin IC upon induction of apoptotic death does not support an active role in chromatin compaction, the down-regulation of nuclear expression levels might suggest a destabilization of chromatin structures in line with the dependence of chromosome movements on nuclear myosin and actin^{45,77-79}.

The new data on the nuclear expression of myosin IC and myosin IIA are now included as new Figure 4 and Supplemental Figure 3 into the revised version of the manuscript, and are included in the results (lines 316-330) and discussion (lines 520-540) sections.

45. Maly, I. V. & Hoffmann, W. A. Myosins in the Nucleus. *Adv. Exp. Med. Biol.* 1239, 213-232 (2020).
46. Cook, A. W., Gough, R. E. & Toseland, C. P. Nuclear myosins-roles for molecular transporters and anchors. *J. Cell Sci.* 133, (2020).
77. Chuang, C. H. et al. Long-Range Directional Movement of an Interphase Chromosome Site. *Curr. Biol.* 16, 825–831 (2006).
78. Hu, Q. et al. Enhancing nuclear receptor-induced transcription requires nuclear motor and LSD1-dependent gene networking in interchromatin granules. *Proc. Natl. Acad. Sci. U. S. A.* 105, 19199–19204 (2008).
79. Mehta, I. S., Amira, M., Harvey, A. J. & Bridger, J. M. Rapid chromosome territory relocation by nuclear motor activity in response to serum removal in primary human fibroblasts. Chromosome positioning dynamics Nuclear myosin 1 β -dependent repositioning of chromosome territories occurs within 15 min-utes of seru. *Genome Biol.* 11, 5 (2010).

b. Is the function of nuclear myosin indeed inhibited by BDM treatment?

Despite the elusive understanding of the functional role, precise regulations and difference across the variety of myosins with a nuclear expression^{45,46}, we have now addressed the effect of the presence of the actomyosin inhibitor BDM during staurosporine-induced apoptosis on nuclear expression levels of myosin IC and myosin IIA.

The results, which are now included as new Fig. 4b,d and e and Suppl Fig 3 b,d and e show that the decrease in expression of the actin-based molecular motor protein myosin IC upon application of

staurosporine was significantly attenuated through concomitant application of BDM during the early phase of apoptosis (2h staurosporine) and late phase of apoptosis (6h staurosporine). In line with no change in nuclear expression levels of myosin IIA during staurosporine-induced apoptosis, expression levels of nuclear myosin IIA also remained unaffected by the additional BDM treatment (Suppl. Fig. 3)

Taken together, we now conclude that the concomitant blockade of actomyosin activity by BDM attenuated changes in nuclear expression of the actin-based molecular motor protein myosin IC, modulated staurosporine-induced chromatin dynamics and resulted in less apoptotic and more necrotic-like cell death.

And discuss this as follows in the revised version of the manuscript:

Discussion 531ff: *The reduction of nuclear myosin IC upon induction of apoptotic death does not support an active role in chromatin compaction, the down-regulation of nuclear expression levels of the unconventional myosin IC, e.g. by redistribution within cellular compartments as described previously⁷⁶, might result in a destabilization of chromatin structures in line with the previously described dependence of nuclear chromosome movements on nuclear myosin and actin⁷⁷⁻⁷⁹. Given the elusive understanding of the functional role, precise regulations and difference across the variety of myosins with a nuclear expression^{45,46}, the refinement of myosin IC and others' explicit roles on chromatin organization under physiological and pathophysiological conditions such as the apoptotic process in neuronal cells remains to be further solved in the future.*

45. Maly, I. V. & Hofmann, W. A. Myosins in the Nucleus. *Adv. Exp. Med. Biol.* 1239, 213-232 (2020).
46. Cook, A. W., Gough, R. E. & Toseland, C. P. Nuclear myosins-roles for molecular transporters and anchors. *J. Cell Sci.* 133, (2020).
76. Maly, I. V. & Hofmann, W. A. Calcium-regulated import of myosin IC into the nucleus. *Cytoskeleton* **73**, 341–350 (2016).
77. Chuang, C. H. *et al.* Long-Range Directional Movement of an Interphase Chromosome Site. *Curr. Biol.* **16**, 825–831 (2006).
78. Hu, Q. *et al.* Enhancing nuclear receptor-induced transcription requires nuclear motor and LSD1-dependent gene networking in interchromatin granules. *Proc. Natl. Acad. Sci. U. S. A.* **105**, 19199–19204 (2008).
79. Mehta, I. S., Amira, M., Harvey, A. J. & Bridger, J. M. Rapid chromosome territory relocation by nuclear motor activity in response to serum removal in primary human fibroblasts Chromosome positioning dynamics Nuclear myosin 1 β -dependent repositioning of chromosome territories occurs within 15 min-utes of seru. *Genome Biol.* **11**, 5 (2010).

c. Do the authors observe the similar results to Figure 2 when nuclear myosin is specifically inhibited?

Upon discussion with experts in the field there is to our knowledge unfortunately no specific nuclear myosin inhibitor available.

5. In Figure 1e-h, "nuclear shrinkage in apoptotic cells" and "swelling in necrotic cells" are defined as time = 0. Are they comparable events? The authors should explain why they defined time = 0 for these.

To quantitatively compare onset, course and sequence of changes in nuclear size, chromatin compaction, i.e. edge count and CCP, and caspase activation before and during apoptotic and non-

apoptotic cell death, **we have temporally aligned the individual recordings of neurons according to the occurrence of a change in nuclear size in Fig. 1.**

Nuclear shrinkage in neurons with an apoptotic fate and swelling in neurons with a non-apoptotic, necrotic like cell fate are comparable in the sense that both are characteristic hallmark features of the respective cell death fate and both processes affect cellular morphology. Only if images of individual neuronal nuclei were temporally aligned according to this unbiased morphological change, we could compare the time course and assess the temporal order of changes in nuclear size, chromatin compaction (edge count and CCP) and caspase activation across neurons of the different fate (control vs. apoptotic vs. necrotic-like cell fate), since the onset of apoptosis or necrosis is different for each neuron. We have added the explanation on this behalf to the results (line 152ff) and the figure legend of Fig. 1.

6. In Figure 1f, when does the edge count start to increase? Was there ever a time when the edge count of these cells was the same as that of the control cells?

For all experiments a baseline period, which lasted at least 20 min before any modification was performed, was recorded, e.g. 0-30 min in Fig. 1 e-h. The acquisition and analysis of this baseline period ensured for example that there was no prior difference across groups, before cells underwent either an apoptotic or non-apoptotic, necrotic-like cell death fate. The absence of differences during this baseline period was also confirmed statistically, i.e. cells during baseline period did not significantly differ across different cell fates (Fig. 1) or pharmacological treatments (Fig. 2). **We have now included the statistical comparison of this baseline period in the revised version of the manuscript and accordingly revised Fig. 1 e-h and Fig. 2 a-d.**

From our observations we noticed that not only the onset of apoptosis upon staurosporine application but also the individual time course of neuronal apoptosis (i.e. the time that a neuron resides in a distinct stage of apoptosis) varied significantly. To compare the average course and sequence of events during the course of apoptotic and non-apoptotic cell death we have aligned corresponding signals of many neurons according to the occurrence of a change in nuclear size in Fig. 1. We have decided to depict a 4-hour time course (2h before and 2h after the apparent change in nuclear size) to highlight in Fig. 1, that chromatin compaction precedes nuclear size change, i.e. significantly increased edge count and CCP in apoptotic and necrotic neurons starting with -2h compared to control. To give a representative example of the kinetics in an individual neuron, we have now included a new supplement figure 4. For the time course of average values and the average onset of the increase in edge count we would like to refer to Fig. 2, where we present the continual course of nuclear size, edge count and CCP of neurons prior to and upon staurosporine application. As described in the results section chromatin compaction, i.e. a significant elevation of average edge count and CCP in staurosporine-treated neurons, can be detected as early as 40 min upon application of staurosporine, which is about 3h before a significant increase in caspase activation is detectable (Results line 234ff).

7. In Figure 4a, all stages should be examined in a same cell. In addition, the stage should be defined using quantitative parameters obtained from multiple cells.

We agree that the super-resolution analysis of the different stages before and during apoptosis of the very same cell would be of high interest. Unfortunately, this is not possible as the **SRM analysis within this study is done on fixed samples and thus precludes a time-lapse analysis of an individual cell.**

But, we have now included quantitative comparison based on confocal, images of different neurons categorized into the different stages (Revised Fig. 5 d-g). We would also refer to Fig. 1, where we show the course of these parameters before and through apoptosis. In addition, **we included a confocal time-lapse images and quantitative analysis of a single representative neuron as a new supplement figure 4.**

8. What is the novel point of *in vivo* analysis? Based on the *in vitro* analysis, the authors should examine the changes in nuclear structure of the cells undergoing apoptosis before Caspase activation.

To our knowledge it is not possible to identify neurons with a prospective caspase activation in tissue. To confirm that chromatin compaction in non-apoptotic neurons and caspase-positive neurons *in vivo*, i.e. in late stages (4 or 5) of apoptosis, share distinct features with the changes we investigated in more depth *in vitro*, we included this SRM data.

Minor points

9. Expression of H2B-mCherry is toxic, so it should be verified by other methods as well.

Primary cortical neurons with overexpression of H2B::mCherry was used by our group in previous studies (Sang et al. 2021 and Peter et al. 2021). In the current study as well as in these previous studies we have carefully checked that the physiological properties (e.g. morphology and electrical profile) are not altered by the overexpression and that also the expression does not affect viability of the neurons. In the present study, we did not notice an increased number of apoptotic cells under control conditions compared to naïve control cultures. We now additionally performed Alamar-based viability assays and did not detect any difference between H2B::mCherry positive vs non-transduced naïve cultures, neither under control conditions nor under staurosporine-induced apoptosis induction. **We have now included the results as new supplement figure 1 and included these results also in the results section of the revised version of the manuscript (line 117ff).**

Relevant references of previous studies of our group with primary cortical cultures and AAV based overexpression of H2B::mCherry:

Wong Fong Sang IE, Schroer J, Halbhuber L, Warm D, Yang JW, Luhmann HJ, Kilb W, Sinning A. Optogenetically Controlled Activity Pattern Determines Survival Rate of Developing Neocortical Neurons. *Int J Mol Sci.* 2021 Jun 19;22(12):6575. doi: 10.3390/ijms22126575.

Peter M, Aschauer DF, Rose R, Sinning A, Grössl F, Kargl D, Kraitsy K, Burkard TR, Luhmann HJ, Haubensak W, Rumpel S. Rapid nucleus-scale reorganization of chromatin in neurons enables transcriptional adaptation for memory consolidation. *PLoS One.* 2021 May 5;16(5):e0244038. doi: 10.1371/journal.pone.0244038.

10. Between which samples does the * in the Figures show the significant difference? What do (condition) or (treatment) mean?

For all statistical comparisons differences across respective experimental groups are indicated. E.g. for figure 1 we applied a mixed-effects analysis and report differences between cells under control conditions vs. apoptotic vs necrotic cells (previously: conditions) and for figure 2 we applied a two-way ANOVA and report differences across pharmacological treatments during baseline and during the time period after application of pharmacological treatments (previously: treatment). Results from multiple comparisons tests are now given in the results section when discussing differences between two specific experimental groups at explicit time points (e.g. 235ff or 247ff).

We have revised the labeling of statistical significance of differences in the figures and included a more detailed explanation of compared experimental groups when applicable (e.g. Fig. 1 and Fig. 2 a-d with legends, Results e.g. 175ff and 245ff).

11. Figure 4b and Figure 6c are meaningless because they quantify the signals in arbitrary and very local area.

The detailed intensity plots presented in Fig. 4b and 6b,c are representative detail views and intensity plots of lines carefully chosen to represent typical distributions of signal intensities. Thus figures illustrate chromatin compaction with super resolution under control (stage 1), early (stage 2) and late phase apoptotic neuronal nuclei (stage 4) *in vitro* and in casp-3 negative and caspase 3 positive cells *in vivo*, respectively. For unbiased systematic comparison this presentation of representative high resolution views is followed by the Voronoi cluster analysis and comparison of radius/density distribution of signals in nuclei of neurons in different apoptotic stages. We would prefer to keep the representative line plots for the sake of sequential readability of the manuscript but have clearly marked these as representative detail views and line plots, when applicable (Fig. 5b and 7b,c with legends).

12. In Discussion, there is no evidence that transcriptional regulation via chromatin structure is important for the significance of the observed changes in nuclear structure prior to Caspase activation. Other possibilities should be discussed.

The discussion was carefully revised and the possibility that other mechanism independent of transcription such as accessibility for macromolecules or robustness of nuclear structure are now included into the discussion (line 483ff).

13. This reviewer do not understand the meaning of the last paragraph of the Discussion. The authors should discuss how this study can contribute to the understanding of "brain function, dysfunction, and development".

The paragraph in the discussion was changed as follows:

Discussion 555 ff: *In conclusion, our study provides evidence for the importance of the tight regulation of chromatin organization in developing neuronal nuclei before and during apoptosis. While chromatin conformation capture techniques revealed that 3D chromatin organization is a fundamental regulatory mechanism in brain function and dysfunction⁵⁸, further investigations of chromatin organization and dynamics in neurons in the context of different physiological and pathophysiological conditions of apoptosis and beyond will improve our understanding of casual relationships between spatial chromatin organization, molecular signatures and neuronal functions.*

58. Maeshima, K., Iida, S. & Tamura, S. Physical Nature of Chromatin in the Nucleus. *Cold Spring Harb. Perspect. Biol.* **13**, a040675 (2021).

Reviewer #2 (Remarks to the Author):

The authors are interested in the underexplored phenomenon of nuclear and chromatin condensation that occurs during cellular apoptosis. To investigate this, they employ confocal microscopy of neurons expressing an H2B-mCherry fusion, along with super-resolution imaging of fixed neurons. They investigate chromatin condensation and nuclear morphology in these cells upon

treatment with staurosporine to induce apoptosis, with or without co-treatment with either an inhibitor of caspase activation or actomyosin activity. Via these techniques the authors divide the process of apoptotic nuclear condensation into 5 stages. Most notably, the earlier stages occur prior to changes in nuclear morphology (e.g. shrinking), and the earliest prior to caspase activation. Caspase inhibition does not inhibit this earliest stage of chromatin condensation, while actomyosin inhibition – here used to prevent the movement of “nuclear bodies” – prevents apoptosis and results in a necrosis-like phenotype.

This study addresses a genuinely underexplored topic in chromatin condensation that accompanies apoptotic cell death. To give some perspective on how glacial progress on this topic has been, a protein termed Acinus was first characterized as playing an important role in apoptotic chromatin condensation in 1999, but the number of research articles dealing with its function in this regard can be counted on the fingers of one hand, and I would argue we still know nothing about what this factor does. So the research topic and the approaches being used here are overdue and most welcome. To a certain extent, the findings reported here raise more questions than they answer, but as an initial foray into this subject I still think this study serves well. There are issues with clarity and possibly logic that I think need to be addressed, however (see specific comments, below).

Comments:

1. In Fig. 2, I'm not sure I see how the caspase inhibitor results in behavior similar to “early granulation observed in staurosporine-only treated cultures.” The graphs show that the similarity runs throughout the time course of the experiment. Is something not clear here? Is the time course shown in Fig. 2 intended to be limited to an “early” phase of apoptosis, and if the authors showed later time points a difference would emerge?

With early chromatin granulation we wanted to emphasize that the chromatin condensation is observed prior to and thus precedes nuclear shrinkage. We agree that the edge count and CCP as quantitative measures of the chromatin compaction develop similarly in presence of staurosporine-only and staurosporine plus caspase inhibitor. This is also supported by a lack of significant differences in time course and extent of early chromatin condensation found between these two treatments in a pair-wise pos-hoc analysis (post-hoc tests 0-6.5h edge count $p \geq 0.35$ and relative CCP $p \geq 0.22$ for staurosporine vs. caspase-3 inhibitor with staurosporine). But treatment with the caspase-3 inhibitor interfered with the staurosporine-induced execution of apoptosis (i.e. blocked caspase activity, strongly reduced nuclear size change and overall apoptotic rate). Thus, we reason that the early increase in chromatin compaction, which temporally precedes caspase activation and nuclear shrinkage, is not affected by pharmacological block of the apoptotic key player caspase-3.

We have **now replaced the term “early” by “preceding” condensation**, when applicable. In addition to the statistical difference across all groups (including also control and staurosporine plus BDM) indicated in the graphs of Fig. 2, **we have now included and discuss all relevant pair-wise post-hoc comparison between relevant groups in the respective results section**. Further, we have included a note to the legend of figure 2, that while the combined application of caspase-3 inhibitor with staurosporine blocked average caspase activation and reduced the nuclear size change, no significant differences in time course and extent of early chromatin condensation were found between caspase-3 inhibitor with staurosporine and staurosporine-only treated neurons.

2. It's a bit of a stretch to go from this experiment to the conclusion that the early chromatin compaction is necessary for apoptosis. This shows that actomyosin is necessary, and proximally perhaps that movement of “nuclear bodies” is also necessary.

Compaction may be the operative principle, but a lot of other possibilities, starting with transcriptional regulation, are still likely (to take one possibility: does the initial stage of compaction simply reflect a global loss of transcription?). The authors do include a disclaimer to this effect, however. Still, it isn't clear what this experiment says about the timing involved – that is, doesn't application of BDM inhibit nuclear reorganization both before and after caspase activation? How do the authors then conclude that the reorganization requirement implied by the BDM experiment applies to the caspase-independent component? At the very least, this point needs clarification.

Blockade of actomyosin activity by BDM resulted in the alteration of staurosporine-induced chromatin dynamics in such a way that the simultaneous application of BDM with staurosporine induced an immediate and unexpected induction of chromatin condensation that lasted throughout the experimental time window of analysis (i.e. 6.5 h). This increase in chromatin condensation occurred earlier than in staurosporine-only treated cells, and was accompanied by a sharp significant increase in nuclear size and suggesting the induction of less apoptotic and more necrotic-like cell death. At the same time, concomitant application of BDM reduced caspase activity by about half (i.e. reduction of NucView-positive cells and caspase activity at 6h in caspase assay), therefore we concluded that changes in chromatin which precede apoptosis are necessary for the execution of the apoptotic cell death in neurons. With the additional experiments addressing the nuclear expression of the conventional and unconventional myosins (specifically myosin IC and IIA) we now show that blockade of actomyosin activity by BDM attenuates staurosporine-induced changes in nuclear expression of the actin-based molecular motor protein myosin IC, modulates staurosporine-induced chromatin dynamics and results in less apoptotic and more necrotic-like cell death. Thereby we substantiated the conclusion that myosin-associated changes in chromatin which precede apoptosis are necessary for the execution of the apoptotic cell death in neurons.

Results and Discussion were revised accordingly.

3. Similarly, I do not understand how the authors conclude that the initial compaction is somehow “not part of the final execution of the apoptotic process per se.” How can they make this distinction? What does this even mean?

The results of the present study show that the additional treatment with caspase-3 inhibitor interfered with the staurosporine-induced execution of apoptosis in neurons (i.e. significant differences in caspase activity between staurosporine-only and staurosporine plus caspase inhibitor in Fig. 2d and e, nuclear size change in Fig. 2a and overall apoptotic rate in Fig. 3b vs 3c), but not with the early chromatin condensation (i.e. no significant difference in immediate increase in edge count and CCP upon staurosporine and staurosporine plus caspase inhibitor treatment in Fig. 2b and 2c). These findings support not only that the early increase in chromatin compaction temporally precedes caspase activation and nuclear shrinkage (see also results Fig. 1) and is thus upstream of caspase-3 activation and of the final execution of the apoptotic process, but also that the chromatin compaction can be experimentally decoupled from the final apoptotic phase by pharmacological block of the apoptotic key player caspase-3. Therefore, we concluded that these early changes in chromatin compaction are not part of the final execution of the apoptotic process per se.

We have included a more detailed explanatory description to this conclusion in the relevant results section (lines 275-282), and hope that the reasoning for this important finding of this study is now clearer.

4. The transition from compaction stages 4 to 5 (Fig. 4) is confusing. What is happening in stage 4? Does this represent discontinuous staining of chromatin within membrane-bounded nuclei or a

fragmentation of the nuclei? How do discrete nuclei then appear again at stage 5? Additional clarification is needed here.

With the description of the progressive stages of apoptosis based on chromatin structure we intended to establish a novel model that describes the chromatin appearance before and during the apoptotic process. In the late phase of apoptosis, i.e. stage 4 and 5, caspase is strongly activated, and the nuclear size is reduced. Before the nucleus enters the final, fully compacted end stage, i.e. stage 5, we observed a prior stage, which is characterized by an already reduced nuclear size, a discontinuous chromatin structure. The quantitative comparison of average size, edge count, H2B::mCherry and NucView signal intensities of neuronal nuclei categorized into stages 1-5 supports the quantitative differences in the measures across the stages and highlights that the main difference between stage 4 and 5 is that increase in H2B::mCherry signal intensity (Revised Figure 5 f). This increase in H2B::mCherry signal intensity suggests that at this stage chromatin is fragmented and at least physically still locally confined to a nuclear region, before the chromatin enters its final fully compacted end stage. Although the nuclear fragmentation is described as a key feature of apoptotic cells, the role of the nuclear envelope and its permeability during cell death is still a matter of current research and subject of recent publications and reviews (e.g. DOI: 10.1038/s41420-020-0256-5).

We have now included the quantitative comparison of average size, edge count, H2B::mCherry and NucView signal intensities of neuronal nuclei categorized into stages 1-5 to the revised Figure 5d-g and included a more elaborated description of the changes we observe during the late phase of apoptosis to the results section of the revised version of the manuscript (lines 364-371).

Reviewer #3 (Remarks to the Author):

Communications Biology/Nature

COMMSBIO-21-2691

The authors described dynamic changes in chromatin organization that lead to the chromatin cluster formations that appear early in developing neurons with a prospective apoptotic fate, although they are not properly part of the final execution of the apoptotic process. Such a chromatin reorganization precedes caspase-3 activation and is distinct from the continuous condensation of the nuclei.

The manuscript is a novel contribution of fundamental importance to the understanding of the apoptosis phenomenon. Overall, it is a major advance in this field. The article is nicely written and technically accurate. It was well planned and executed. Sophisticated chromatin conformation capture techniques were used for the evaluation of the preparations. The results are well described and discussed with adequate literature survey. Although I have no expertise in the Voronoi analysis and respective statistics I can understand the importance of the authors' contribution.

We thank reviewer 3 for the review of the manuscript and have now applied suggested changes according to the minor comments in the revised version of the manuscript.

Minor:

- Many sequential words and symbols in the text appear frequently not separated by space. Ex.: 163nm, was490, etc.

- Materials and Methods

line 632 – Please write AraC in full the first time its mentioned (cytarabine).

line 636 – Please write AAV in full.

line 718 – Remove “chromatin condensation parameter”. CCP has been defined in the preceding line. Standardize: minutes or min; hours or h?

- Results

line 114 – “nucleus” swelling Fig. 1.

For easier reader identification of relative time in e and f, please insert values in the X line

References line 918 – abbreviate the name of the journal

Revised and new figures :

Revised Figure 1. Nuclear structure and chromatin dynamics during apoptotic and non-apoptotic, necrotic-like cell death in primary cortical neurons. **a)** Representative primary cortical culture at DIV 8 transduced with recombinant **adeno-associated virus (AAV)** to express the chromatin marker H2B::mCherry under the human synapsin promoter; left: phase contrast, middle: H2B::mCherry, right: merge; scale 100µm. Detail on the far right: top: merged, bottom: H2B::mCherry; scale 10µm.

b) Hourly captures of representative cells during a 7h time lapse acquisition, before and after start of apoptotic or non-apoptotic processes, scale 10 μ m. Gray vertical lines indicate which time points are depicted in (c), see **SupMov.1-5**. **c)** Depiction of chromatin dynamics during the most striking nuclear changes for typical types of cell death between time points indicated with the gray vertical lines in (b). **d)** Typical examples for raw confocal signal (top) and Sobel-filter-based edge detection (bottom) under control conditions (left) and apoptotic neurons (right), scale 10 μ m. **e-h)** Aligned and normalized average nuclear size, edge count, chromatin condensation parameter (CCP), and caspase activity normalized to baseline for representative neurons surviving under control conditions (black, n= 11), classified as apoptotic (red, n= 29) and classified as non-apoptotic, necrotic-like (gray, n= 19). Corresponding signals were aligned according to the occurrence of a change in nuclear size (t= 0h is time of shrinkage for apoptosis and swelling for non-apoptosis, necrosis-like cell death). Imaging was performed every 10min and the first 3 time points correspond to baseline. Data are represented as mean \pm SEM. For e-h) mixed-effects analyses for differences between cells with apoptotic vs necrotic cell fate vs. control conditions were applied during baseline: nuclear size $F(2, 168)=1.51e^{-15}$, $p>0.9999$; edge count $F(2, 168)=5.48e^{-16}$, $p>0.9999$; CCP $F(2, 168)=3.53e^{-16}$, $p>0.9999$; caspase activity $F(2, 168)=1.01e^{-11}$, $p>0.9999$, and for the aligned time period of -2 to +2h: nuclear size $F(2, 56)=107.7$, $p<0.0001$; edge count $F(2, 56)=10.82$, $p=0.0001$; CCP $F(2, 56)=8.87$, $p=0.001$; caspase activity $F(2, 56)=23.33$, $p<0.0001$.

Revised Figure 2. Pharmacological inhibition of caspase-3 blocked apoptosis but not preceding condensation, whereas blockade of myosin activity induced non-apoptosis, necrosis-like cell death. Live cell confocal images were acquired every 10min for 3 time points before the application of the pharmacological treatments (baseline), and for an additional 39 time points (i.e. 6.5h) after application of indicated pharmacological treatments. **a - c)** Normalized nuclear size, edge count and CCP for all neurons under control (black, n= 460 cells), staurosporine (1.5 μ M; orange, n= 518 cells), BDM (20mM) with staurosporine (blue, n= 301 cells), and caspase-3 inhibitor Z-DEVD-FMK (100 μ M) with staurosporine (green, n= 265 cells). **Note, that while the combined application of caspase-3 inhibitor with staurosporine blocked average caspase activation and reduced the nuclear size change, no significant differences in time course and extent of early chromatin condensation were found between caspase-3 inhibitor with staurosporine and staurosporine-only treated neurons.** **d)** Mean percentage of NucView-positive cells (caspase activation) per culture (control n= 9 cultures, staurosporine n= 9 cultures, caspase-3 inhibitor Z-DEVD-FMK with staurosporine n= 5 cultures, BDM

with staurosporine n= 5 cultures) identified throughout the acquisition. **e)** Caspase activity measured by luminescent caspase assay at DIV8 under baseline conditions, as well as 3h and 6h after start of pharmacological treatments. Data are represented as mean \pm SEM. For a-d) two-way ANOVA was applied to detect differences across pharmacological treatments during baseline: nuclear size $F(3, 4620)=0$, $p>0.9999$; edge count $F(3, 4620)=3.45e^{-13}$, $p>0.9999$; CCP $F(3, 4620)=0.0$, $p>0.9999$; mean percentage of NucView-positive cells $F(3, 24)=0.000$, and during the time period after application of pharmacological treatments: nuclear size $F(3, 60060)=906.2$, $p< 0.0001$; edge count $F(3, 60060)=1526$, $p<0.0001$; CCP $F(3, 60060)=2344$, $p<0.0001$; mean percentage of NucView-positive cells $F(3, 24)=4.82$, $p=0.0092$. For e) a Kruskal-Wallis test was applied: $H(7)=39.56$, $p<0.001$.

New Figure 4. Expression of the unconventional myosin IC in nuclei of primary cortical neurons decreases with the onset of apoptosis, and concomitant application of the actomyosin inhibitor BDM attenuates this decrease induced by staurosporine treatment. a) Representative confocal images of immunostainings against myosin IC, respective H2B::mCherry signals and merged images

with horizontal z-stack projections confirm the expression of myosin IC in the nucleus of cortical neurons with a mostly complementary expression of myosin IC and H2B::mCherry under control conditions. In the early and late phase of apoptosis nuclear myosin IC signal decreased continuously (scales 10 μ m). **b)** Representative average z-stack projections of myosin IC signals under untreated control conditions, 2h and 6h after staurosporine application and upon additional treatment with the actomyosin inhibitor BDM (scale 20 μ m). **c)** Nuclear myosin IC signal intensity significantly decreased upon application of staurosporine for 2h and 6h (n=33/48/33 cells). **d,e)** The concomitant application of the actomyosin inhibitor BDM significantly attenuated the staurosporine-induced decrease in nuclear myosin IC intensity, both after 2h (n=48/19 cells) and after 6h (n=33/19 cells). Data are represented as boxplots, whiskers MIN to MAX. One-way ANOVA was applied for comparison of differences between control and staurosporine 2h and 6h $F(2, 123) = 46,73$ $p < 0,0001$ and t-test for comparison of nuclear myosin IC signal intensity upon application of staurosporine only with staurosporine plus BDM after 2h ($p < 0,0001$) and 6h ($p < 0,05$), respectively.

Revised Figure 5. Classification of five progressive stages of apoptosis based on chromatin structure. **a**) Live confocal images (left, time-stamped images) followed by SMLM (images with typical details highlighted by white boxes) after fixation of the same nucleus from a cultured primary cortical neuron expressing H2B::mCherry; scale: $5\mu\text{m}$ in both. Time stamps on the confocal images refer to time elapsed since the beginning of the acquisition. Note, in cells identified to be at stages 4 and 5, a transient caspase-3 activation (indicated by NucView*) was detectable during the confocal acquisition. **Representative** details highlighted in super resolution images (white boxes, scale $0.5\mu\text{m}$) show the condensation of chromatin increasing from stage 1 to stage 5. **b**) **Intensity plots** of lines in

representative detail views for stages 1, 2, and 4 show a stepwise increase in signal intensity, which indicates an increase in the condensation of chromatin clusters in neuronal nuclei. **c)** Theoretical model describing chromatin appearance during the five identified stages of apoptosis based on the observed morphological changes presented by neuronal nuclei before and during the apoptotic process. **d-g)** Quantitative comparison of average nuclear size, edge count, H2B::mCherry and NucView signal intensities of neuronal nuclei categorized into stages 1-5 (n=13/6/4/3/2 cells). Data are represented as mean \pm SEM. One-way ANOVA was applied for comparison of nuclear size $F(4, 23)=27.02$ $p<0.0001$ and edge count $F(4, 23)=4.11$ $p=0.0118$, Kruskal-Wallis test for comparison of H2B::mCherry signal intensity $H(5)=12.49$, $p=0.0141$, and caspase activation $H(5)=16.75$, $p=0.002$ across nuclei at different stages.

Revised Figure 6. Voronoi analysis of H2B::mCherry signal localizations of super resolution data confirms a progressive condensation of chromatin before and during the apoptotic process. a) Schematic illustration of Voronoi tessellation used for calculation of Voronoi densities based on single molecule localizations. **b)** Distribution of the log-normalized Voronoi densities representing the density of H2B::mCherry molecules within nuclei, grouped by apoptotic stage. To account for different number of localizations and cells for each stage, the area under the curve is normalized to 1 and the probability density function is plotted; a right-shift of the peak represents an increase in density of individual H2B::mCherry localizations, thus chromatin condensation. Stage 1: blue (n= 13 cells), stage 2: orange (n= 6 cells), stage 3: green (n= 4 cells), stage 4: red (n= 3 cells), stage 5: purple (n= 2 cells). **b)** For the calculation of radius by density distribution, signal positions were transformed from a cartesian to a polar coordinate system for each nucleus. Respective radii from the center of mass of the nuclei were calculated for individual localizations (in nm) and related to the Voronoi density. The values of the normalized (area under curve = 1) probability density function are represented by colors (see color bar on the right).

Revised Figure 7. Super resolution SMLM imaging of apoptotic and non-apoptotic neurons in mouse cortical cryosections confirms high chromatin condensation during apoptosis *in vivo*. **a)** Widefield imaging of aCasp3 immunosignal (left) allowed the identification of apoptotic and non-apoptotic neurons, and SMLM reconstruction of DNA molecules stained with Sytox Orange (right) are shown for four representative cells imaged in cortical tissue samples from a P6 mouse. Scale: 5µm. **b)** **Representative** detailed views on inserts highlighted in (a) revealed that condensation of chromatin into clusters is higher in nuclei of neurons with aCasp3 signal (red) as compared to nuclei without caspase 3 activation (blue). Scale 0.5µm. **c)** Intensity plots of **representative lines** from details in (b) confirmed the higher condensation of chromatin in nuclei of neurons with the aCasp3 signal. **d)** Distribution of the log-normalized Voronoi densities of aCasp3-positive (red, n= 6 cells) and -negative neurons (blue, n= 7 cells). To account for different number of localizations and cells under each condition, the area under the curve is normalized to 1 and the probability density function is plotted.

New Supplemental Figure 1. Normal viability of primary cortical neurons with H2B::mCherry overexpression under control conditions and no difference in viability under staurosporine-treatment compared to neurons without H2B::mCherry. **a)** Alamar-based assay did not show a significant difference in viability when comparing viability of No H2B::mCherry vs H2B::mCherry neurons. 4h treatment with staurosporine (1.5µM) resulted in a similar reduction of neuronal viability. Data are represented as mean ± SEM. Two-way ANOVA was applied for comparison of differences between control and staurosporine $F(1, 21) = 31,34$ $p < 0.0001$ and No H2B::mCherry vs. H2B::mCherry $F(1, 21) = 0,1335$ and $p = 0.72$.

New Supplemental Figure 2. Co-localization of heterochromatin marker H3K27me3 with H2B::mCherry fluorescence signal. a) Representative confocal images of nuclei of a neuron under control condition and upon induction of apoptosis with staurosporine (scale 5 μ m). **b)** High average Pearson's Coefficients show strong correlation of immunohistochemical signal of anti-H3K27me3 staining with the transgenic H2B::mCherry signal under control conditions (later defined as stage 1, n=5 cells) and during all phases of apoptosis (stage 2-5, n=7 cells).

New Supplementary Figure 3. Myosin IIA is expressed in the nucleus of primary cortical neurons at similar levels under control and apoptotic conditions and nuclear expression levels are not affected by the additional application of BDM. a) Confocal images of immunostainings against myosin IIA and

H2B::mCherry signal confirm the expression of myosin IIA in the nucleus of cortical neurons both under control conditions and in apoptotic neurons (scale 10 μ m). **b**) Representative average z-stack projections of myosin IIA signals under untreated control conditions, 2h and 6h after of staurosporine application and upon additional treatment with the actomyosin inhibitor BDM (scale 20 μ m). **c**) Nuclear myosin IIA signal intensity was not significantly altered by application of staurosporine (n=13/11/11 cells). **d,e**) The additional application of BDM did not affect the nuclear myosin IIA signal intensity neither after 2h nor 6h (n = 11 cells per condition). Data are represented as boxplots, whiskers MIN to MAX. One-way ANOVA was applied for comparison of differences between control and staurosporine 2h and 6h $F(2, 32) = 0.37$ $p=0.70$ and t-test for comparison of nuclear myosin IIA signal intensity upon application of staurosporine only with staurosporine plus BDM after 2h ($p=0.24$), and 6h ($p=0.10$).

New Supplementary Figure 4. High-resolution, confocal time-lapse imaging (a) and quantitative analysis (b) of a single, representative neuronal nucleus before, during and after apoptosis confirms that the increase in chromatin compaction precedes the decrease in nuclear size and caspase activation. Confocal images were acquired every 10min, starting before the apoptosis process began until the end of apoptosis. Already before nuclear shrinkage (I, from 40 min onwards), chromatin becomes more compacted throughout apoptosis, i.e. increase in Edge count (II) and CCP (III) starting to rise with 20 min. In line with the continuous shrinkage of the nucleus, the H2B::mCherry signal becomes more intense from 40-50 min on (IV). Caspase activation, visualized by the NucView signal (V) follows with a delay in time at around 70-80 min, where after it slowly fades.

Reviewers' comments:

Reviewer #1 (Remarks to the Author):

The authors have addressed most of the criticisms and performed significant experiments. This reviewer respects their effort, especially for the experiments of myosins. However, the manuscript contains one serious concern as follows.

1. The authors explained the relationship between the phenomenon they observed and chromatin condensation/compaction in the rebuttal letter. However, this reviewer is still not convinced by their explanation. This is because their observation, such as the increase in edges or shift in Voronoi value using microscopy, does not necessarily involve chromatin condensation but only chromatin aggregation. A system similar to their observation is SAHF formation during cellular senescence, in which DAPI foci increase and presumably Sobel edges also increase during SAHF formation. However, no increase in heterochromatin domains is observed during SAHF formation (Chandra, Mol Cell, 2012). This suggests that there is a context showing the changes in chromatin structure by DNA staining without chromatin condensation. It is possible that the authors' results can be similar. Therefore, the use of terms "chromatin condensation," "chromatin compaction," or "CCP" without explicit definition is very misleading. The reviewer strongly requests either the following two improvements.

- Examining the chromatin condensation by other methods, such as ATACsee and FRET. ATACsee (not ATAC-seq) and FRET are methods to examine chromatin condensation on a single cell level using microscopy and are fully applicable to the authors' system.
- Replacing all of the terms "chromatin condensation" (including CCP) in the manuscript into the appropriate term, such as "chromatin structural change." "Chromatin compaction" may also be appropriate if they clearly define it and distinguish it from the changes in chromatin accessibility.

Minor comments

2. The following sentence needs a reference.

page 5, line 153, "Nuclear shrinkage is a typical hallmark for neurons with an apoptotic fate, whereas non-apoptotic, necrotic-like cell death is characterized by nuclear swelling."

3. It is difficult to discuss that the chromatin structural change preceding apoptosis is the "upstream" of caspase-3 activation from their results, as described on page 7, line 279, and page 12, line 492. The authors should rephrase them.

4. In the single-molecule imaging, this reviewer is wondering whether the efficiency of blinking of the fluorescence is comparable between non-apoptotic and apoptotic cells since the intracellular condition, such as pH, is different. Did the authors check it?

Reviewer #2 (Remarks to the Author):

In response to reviewer comments, the authors have made extensive edits and revisions to the text and figures, including new figures with additional data. They have thus been highly responsive to criticisms raised regarding the first version of this manuscript. In most cases I think they have answered the questions of clarity that I brought up. I'm afraid I'm mystified by their response to my original comment 2 ("It's a bit of a stretch..."), which was concerned with two issues: (1) the "early" (now re-phrased as "preceding") stage of chromatin compaction may simply reflect a loss of transcriptional activity or other influence; and (2) that the BDM experiments do not necessarily demonstrate that "early" compaction is, in fact, necessary for apoptotic cell death. The first issue is not a large one, insofar as it involves mostly speculation.

For the second issue, however, the authors make a counter-argument that seems to rely on partial loss of caspase activity upon treatment with BDM, although I'm not 100% certain of this interpretation, since their argument is difficult to follow here. This may represent a language issue (?).

Assuming that I have sufficient understanding of what they have written here, I am not convinced that necrotic cell death that appears to ensue upon simultaneous application of BDM and staurosporine can be taken to signify anything regarding the pre-apoptotic (e.g. pre-caspase activation) and apoptotic mechanisms at work normally. The necrotic phenotype is very different, involving an immediate and dramatic condensation of chromatin concomitant with an increase in cell size (as the authors have noted, several times now). When BDM "attenuates" or "modulates" staurosporine-induced responses, it is doing so in the context of those responses, and it is therefore not clear that the "preceding" chromatin condensation or lack thereof have anything to do with this.

To be more concise: if the model is that "early" chromatin condensation is necessary for later execution of the apoptotic pathway, I don't see how the BDM experiments demonstrate this, when BDM is present in the later stages as well and could therefore be exerting its primary effects on apoptosis then.

Regardless, my original view of the manuscript – that it represents a potentially valuable initial foray into the question of chromatin condensation during apoptosis – still stands, and the revised manuscript represents an improvement over the first version.

Reviewer #3 (Remarks to the Author):

From my point of view, and after considering how the authors replied to all the referees' comments, I concluded that the authors did a very good job in revising their manuscript.

Reviewers' comments:

Reviewer #1 (Remarks to the Author):

The authors have addressed most of the criticisms and performed significant experiments. This reviewer respects their effort, especially for the experiments of myosins. However, the manuscript contains one serious concern as follows.

1. The authors explained the relationship between the phenomenon they observed and chromatin condensation/compaction in the rebuttal letter. However, this reviewer is still not convinced by their explanation. This is because their observation, such as the increase in edges or shift in Voronoi value using microscopy, does not necessarily involve chromatin condensation but only chromatin aggregation. A system similar to their observation is SAHF formation during cellular senescence, in which DAPI foci increase and presumably Sobel edges also increase during SAHF formation. However, no increase in heterochromatin domains is observed during SAHF formation (Chandra, Mol Cell, 2012). This suggests that there is a context showing the changes in chromatin structure by DNA staining without chromatin condensation. It is possible that the authors' results can be similar. Therefore, the use of terms "chromatin condensation," "chromatin compaction," or "CCP" without explicit definition is very misleading. The reviewer strongly requests either the following two improvements.

a. Examining the chromatin condensation by other methods, such as ATAC-seq and FRET. ATAC-seq (not ATAC-seq) and FRET are methods to examine chromatin condensation on a single cell level using microscopy and are fully applicable to the authors' system.

b. Replacing all of the terms "chromatin condensation" (including CCP) in the manuscript into the appropriate term, such as "chromatin structural change." "Chromatin compaction" may also be appropriate if they clearly define it and distinguish it from the changes in chromatin accessibility.

Changes in histone structures, assessed in the present study by imaging of nuclear H2B::mCherry signal density, can reflect changes in either chromatin compaction or condensation (Irianto et al., 2014). The latter is from our point of view more likely and supported by the high correlation of the H2B::mCherry signal with the heterochromatin marker trimethyl-Histone H3, both under control and apoptotic conditions (Suppl. Fig. 2), as well as results from a previous study in which very similar changes upon activity modulation could be linked to strong transcriptional changes (Peter et al 2021). But, we agree with Reviewer 1 that we cannot exclude that the observed structural changes in chromatin structure during apoptosis are at least partly attributed to structural rearrangement without chromatin condensation as described previously e.g. in senescent cells (Chandra et al., 2012). **Thus, we have decided that we now throughout the manuscript refer to the observed increase in H2B::mCherry signal densities as chromatin compaction. Therefore, have replaced "chromatin condensation" (including CCP) to "chromatin compaction" and redefined CCP as chromatin compaction parameter.** In addition, we have included a more elaborated explanation in the introduction, results and discussion section as follows:

Introduction page 3 lines 91ff

Thus, the level of compaction of chromatin is related to silencing/activation of chromosomal territories^{27,28} and has also experimentally been shown to influence gene expression under different physiological and pathological conditions^{14,16,29}. But, structural rearrangements

without chromatin condensation have been described previously e.g. the formation of senescence-associated heterochromatic foci³⁰. Thus, structural changes in chromatin such as an increase in chromatin compaction have to be interpreted with caution and cannot be directly linked to changes in chromatin accessibility. For quantification of structural compaction and condensation of chromatin, Sobel edge detection algorithms can be used, since both compaction and condensation of chromatin increases the number of distinct spaces defined as edges within the nucleus²⁷.

Results page 4 lines 127ff

Changes in H2B::mCherry signal densities within the nucleus of transduced neurons reflect changes in chromatin compaction or condensation, both are generally associated with an increase of spaces²⁷. Consistent with an increase in chromatin condensation, H2B::mCherry fluorescence within nuclei of neurons was strongly co-localized with the heterochromatin marker trimethyl-Histone H3 (Lys27)³², both under control conditions and during the apoptotic process (Suppl. Fig 2, Pearson's Coefficient: control 0.93 ± 0.004 apoptosis 0.86 ± 0.02 , $n=5/7$ cells). Yet, it cannot be excluded that the observed structural changes in chromatin structure during apoptosis are at least partly attributed to structural rearrangement without chromatin condensation as described previously e.g. in senescent cells³⁰, thus we refer to the increase in H2B::mCherry signal densities as chromatin compaction.

Discussion page 13 lines 516ff

Independent from their physical nature, and based on the overwhelming evidence, it seems to be clear that changes in chromatin compaction are in most cases associated with changes in chromatin condensation and thus affect transcription in multiple cellular processes. With few exceptions³⁰, changes in chromatin compaction thus likely resemble the structural correlate of transcriptional changes e.g. during the early apoptotic phase in cortical neurons as shown in the present study. In developing neurons, the prominent apoptotic pathway is believed to be the intrinsic (mitochondrial) apoptotic pathway^{70,71}, where activation of transcriptional programs is involved in the execution of the mitochondrial apoptotic pathway before caspase 3 is activated. In this scenario, chromatin compaction and condensation processes would allow for transcription of pro-apoptotic genes, whose protein products lead to the initiation of cascade events, including the pro-apoptotic, pore-forming BAX and BAK proteins, from the BCL-2 family of proteins⁷² and finally the apoptotic deconstruction of the designated neurons. In line, very similar, but transient structural changes could be directly associated with an activity-dependent increase in gene expression of immediate early genes and a transcriptional activation in a previous study²¹. However, the structural increase in chromatin compaction in apoptotic neurons could in principle also reflect a general loss of transcriptional activity. In the future, combined high resolution structural and functional analysis is needed to fully understand the functional consequences of structural changes at the molecular level. Thus, we can only speculate that the transcription of genes that lead to the execution of apoptosis is enabled by the increased chromatin condensation preceding caspase-3 activation and major changes in nuclear structure.

Minor comments

2. The following sentence needs a reference. page 5, line 153, "Nuclear shrinkage is a typical hallmark for neurons with an apoptotic fate, whereas non-apoptotic, necrotic-like cell death is characterized by nuclear swelling."

References were included

3. It is difficult to discuss that the chromatin structural change preceding apoptosis is the "upstream" of caspase-3 activation from their results, as described on page 7, line 279, and page 12, line 492. The authors should rephrase them.

In the results section (page 8 line 287 and page 12 line 500), we have replaced the term "upstream" by descriptions of the respective temporal order of events, i.e. nuclear compaction precedes or antecedes caspase 3 activation which is the direct experimental conclusion of the results.

4. In the single-molecule imaging, this reviewer is wondering whether the efficiency of blinking of the fluorescence is comparable between non-apoptotic and apoptotic cells since the intracellular condition, such as pH, is different. Did the authors check it?

For super-resolution microscopy, specimens were fixed by paraformaldehyde and subsequently exposed to different washing, staining and imaging solutions to allow the immunohistochemically detection e.g. of activated caspase 3 on the one hand and the imaging of single fluorescence molecules on the other hand. Therefore, β -Mercaptoethanol- or fBALM induced quenching of mCherry or SytoxOrange is very unlikely to be affected by prior differences in intracellular milieu. To account for variations in the number of signal localizations across different neurons and stages the probability density functions of Voronoi density histograms for SRM analysis were calculated and normalized (such that the area under the histogram is set to 1) as described in the methods (Page 29 Line 925ff).

Reviewer #2 (Remarks to the Author):

In response to reviewer comments, the authors have made extensive edits and revisions to the text and figures, including new figures with additional data. They have thus been highly responsive to criticisms raised regarding the first version of this manuscript. In most cases I think they have answered the questions of clarity that I brought up.

I'm afraid I'm mystified by their response to my original comment 2 ("It's a bit of a stretch..."), which was concerned with two issues: (1) the "early" (now re-phrased as "preceding") stage of chromatin compaction may simply reflect a loss of transcriptional activity or other influence; and (2) that the BDM experiments do not necessarily demonstrate that "early" compaction is, in fact, necessary for apoptotic cell death. The first issue is not a large one, insofar as it involves mostly speculation.

For the second issue, however, the authors make a counter-argument that seems to rely on partial loss of caspase activity upon treatment with BDM, although I'm not 100% certain of this interpretation, since their argument is difficult to follow here. This may represent a language issue (?). Assuming that I have sufficient understanding of what they have written here, I am not convinced that necrotic cell death that appears to ensue upon simultaneous application of BDM and staurosporine can be taken to signify anything regarding the pre-apoptotic (e.g. pre-caspase activation) and apoptotic mechanisms at work normally. The necrotic phenotype is very different, involving an immediate and dramatic condensation of chromatin concomitant with an increase in cell size (as the authors have noted, several times now). When BDM "attenuates" or "modulates" staurosporine-induced responses, it is doing so in the context of those responses, and it is therefore not clear that the "preceding" chromatin condensation or lack thereof have anything to do with this.

To be more concise: if the model is that "early" chromatin condensation is necessary for later

execution of the apoptotic pathway, I don't see how the BDM experiments demonstrate this, when BDM is present in the later stages as well and could therefore be exerting its primary effects on apoptosis then.

Regardless, my original view of the manuscript – that it represents a potentially valuable initial foray into the question of chromatin condensation during apoptosis – still stands, and the revised manuscript represents an improvement over the first version.

We thank Reviewer 2 for his careful and elaborated explanation of the open issues.

We have now included the possibility that the increase in chromatin compaction might in principle also reflect a loss in transcriptional activity to the discussion of the results (page 13 line 531f).

For the interpretation of the BDM experiments we have now included additional data of experiments, in which BDM was only present in the early phase of apoptosis (i.e. 0-2 h upon staurosporine application) and not in the later phase (i.e. 2-6 h upon staurosporine application). The results show that the application of BDM during the early phase is sufficient to significantly mitigate the staurosporine-induced activation of caspases. Thus, these new results support that acto-myosin activity, which alters chromatin remodeling during the early phase of apoptosis, is essential for apoptotic death in immature cortical neurons as previously discussed in the manuscript.

The Results are included as follows in the results section of the manuscript (page 9 line 339ff):

To assess whether actomyosin activity in the early phase of apoptosis induction is critical for the execution of apoptosis, we applied BDM only during the first 2 hours of staurosporine application and analyzed the caspase activation after an additional 4 hours of only staurosporine treatment, i.e. 6h. The presence of BDM during the early phase (0-2h) significantly attenuated the staurosporine-induced activation of caspase (control $100 \pm 3.59\%$, staurosporine $158.20 \pm 6.86\%$, staurosporine with BDM 0-2h $126.2 \pm 10.60\%$; $n=6$ cultures each; post-hoc test $p=0.02$ for staurosporine vs. staurosporine with BDM 0-2h, $p=0.07$ for control vs. staurosporine with BDM 0-2h; Suppl. Fig. 4).

And were also added as new Supplemental Figure 4:

Supplementary Figure 4. The presence of BDM during the early phase (0-2h) mitigates the staurosporine-induced caspase activation after 6h. Caspase activity measured by luminescent caspase assay at DIV8 or DIV9 under untreated control conditions, as well as upon staurosporine treatment for 6h as well as staurosporine treatment for 6h with BDM application from 0-2h. Data are represented as mean \pm SEM. $n=6$ cultures per condition. One-way ANOVA was applied to detect differences across pharmacological treatments $F(2,15)=14.78$ and $p=0.0003$ and results from subsequent Tukey's multiple comparison test are shown.

Reviewer #3 (Remarks to the Author):

From my point of view, and after considering how the authors replied to all the referees' comments, I concluded that the authors did a very good job in revising their manuscript.

REVIEWERS' COMMENTS:

Reviewer #1 (Remarks to the Author):

All the points have been properly addressed by the authors. This reviewer respects their great efforts.

Reviewer #2 (Remarks to the Author):

The authors have responded to all of my own concerns.